# Long-read transcript sequencing identifies differential isoform expression in the entorhinal cortex in a transgenic model of tau pathology

Szi Kay Leung ®[1] ✉, Rosemary A. Bamford[1], Aaron R. Jeffries ®[2], Isabel Castanho ®[1,3,4], Barry Chioza ®[1], Christine S. Flaxman[1], Karen Moore ®[2], Emma L. Dempster ®[1], Joshua Harvey ®[1], Jonathan T. Brown[1], Zeshan Ahmed[5], Paul O'Neill[2], Sarah J. Richardson ®[1], Eilis Hannon ®[1] & Jonathan Mill ®[1] ✉

Increasing evidence suggests that alternative splicing plays an important role in Alzheimer's disease (AD) pathology. We used long-read sequencing in combination with a novel bioinformatics tool (*FICLE*) to profile transcript diversity in the entorhinal cortex of female transgenic (TG) mice harboring a mutant form of human tau. Our analyses revealed hundreds of novel isoforms and identified differentially expressed transcripts – including specific isoforms of *Apoe*, *App*, *Cd33*, *Clu*, *Fyn* and *Trem2* – associated with the development of tau pathology in TG mice. Subsequent profiling of the human cortex from AD individuals and controls revealed similar patterns of transcript diversity, including the upregulation of the dominant *TREM2* isoform in AD paralleling the increased expression of the homologous transcript in TG mice. Our results highlight the importance of differential transcript usage, even in the absence of gene-level expression alterations, as a mechanism underpinning gene regulation in the development of AD neuropathology.

Alzheimer's disease (AD) is a devastating neurodegenerative disorder that is clinically characterized by progressive memory loss, cognitive decline, and behavioral impairment[1]. These symptoms result from brain atrophy, synaptic loss and progressive neuropathology involving the extracellular accumulation of β-amyloid (Aβ) proteins in amyloid plaques and the intracellular aggregation of hyperphosphorylated tau into neurofibrillary tangles[2]. Despite recent success in identifying genetic risk factors for AD, the mechanisms driving disease progression remain unclear.

Alternative splicing (AS) is a post-transcriptional regulatory mechanism that generates multiple RNA isoforms from a single transcribed messenger RNA (mRNA) precursor. By dramatically increasing transcriptional diversity from the coding genome, AS represents an important regulator of gene expression. Transcriptional dysregulation, and in particular mis-splicing, has been shown to play a key role in the development and pathogenesis of AD[3,4] and transgenic mouse models of AD pathology[5,6]. Hundreds of genes have been reported to be differentially spliced in post-mortem brain tissue from AD patients, with the differential expression of specific transcripts of *APOE*, *BIN1* and *APP*[3]. Previous studies investigating AS have been limited by the use of short-read RNA sequencing (RNA-Seq) approaches, which cannot span full-length transcripts and unequivocally detect specific isoforms[7]. Recent advances in long-read sequencing have addressed these challenges; Pacific Biosciences (PacBio) isoform sequencing (Iso-

[1]Department of Clinical and Biomedical Sciences, University of Exeter, Exeter, UK. [2]Biosciences, University of Exeter, Exeter, UK. [3]Department of Pathology, Beth Israel Deaconess Medical Center, Boston, MA, USA. [4]Harvard Medical School, Boston, MA, USA. [5]Eli Lilly, Cambridge, MA, USA.
✉e-mail: s.k.leung@exeter.ac.uk; j.mill@exeter.ac.uk

Seq) and Oxford Nanopore Technologies (ONT) nanopore sequencing can generate long reads (>10 kb), enabling the direct assessment of alternatively spliced transcripts[8]. We recently demonstrated the power of long-read sequencing for the characterization of transcript diversity and quantification in the human and mouse cortex[9].

In this study, we used PacBio Iso-Seq and ONT nanopore cDNA sequencing to examine isoform diversity and AS dysregulation in a well-characterized transgenic model of tau pathology[10], rTg4510 (Fig. 1A). This mouse model over-expresses a human mutant (P301L) form of the microtubule-associated protein tau (MAPT) and is characterized by widespread transcriptional dysregulation paralleling the accumulation of tau pathology in the cortex[10]. The spread of tau pathology in rTg4510 mice[5] closely recapitulates the Braak stages in human AD[11], with appearance of pre-tangles from as early as 3 months followed by synaptic and neuronal loss by 9 months. We previously identified widespread differences in gene expression associated with the development of tau pathology in rTg4510 mice[5], although differences at the transcript level were not explored due to the limitations of short-read RNA-Seq. In this study, we first performed whole transcriptome long-read sequencing of the entorhinal cortex (EC) − a brain region defined by early neuropathology in AD[11]− profiling tissue from both wild-type (WT) and transgenic (TG) mice at two time points (2 and 8 months) that represent early and late stages of tauopathy[10]. Second, we performed ultra-deep targeted long-read sequencing of transcripts expressed from 20 genes previously implicated in AD and other neurodegenerative disorders (Table 1) in tissue from WT and TG mice at four time points (2, 4, 6 and 8 months) to identify differentially expressed transcripts (DETs) in the EC associated with tau pathology progression. Third, we used fluorescence-activated nuclei sorting (FANS)[12] followed by ONT nanopore sequencing to explore the extent to which DETs associated with tau pathology are expressed in specific cell populations. Finally, to explore the relevance of our findings to AD, we also performed targeted ONT sequencing in post-mortem human cortex samples from individuals with and without advanced AD neuropathology. As part of this study, following the implementation of our customized analysis pipeline (Fig. 1B), we developed a novel bioinformatics tool − **F**ull **I**soform **C**haracterisation from **L**ong-read sequencing **E**xperiments (*FICLE*) − that facilitates the accurate assessment of AS events and the visualization of isoforms based on splicing patterns (Fig. 1C), leveraging the high sequencing coverage of targeted long-read transcriptome datasets. Our transcript annotations and data analysis pipeline are available as a resource to the research community (see Data Availability). Our results confirm the importance of AS in mediating transcript diversity in the cortex and provide evidence of differential transcript usage, even in the absence of gene-level expression alterations, associated with the development of tau pathology in TG mice.

## Results

### Whole transcriptome long-read sequencing identifies specific transcripts of Gfap associated with the progression of tau pathology

We previously used short-read RNA-Seq to identify changes in EC gene expression associated with the development of tau pathology in TG mice carrying a human mutant form of the *MAPT* gene[5]. In the current study, we used long-read cDNA sequencing to extend these analyses and profile alternative splicing and differential transcript expression in the same mice. Our first analyses used PacBio Iso-Seq to generate whole transcriptome long-read sequencing data from a subset of WT and TG mice ($n = 12$, 3 WT and 3 TG at ages 2 and 8 months) (Fig. 1A, Supplementary Data 1) with raw reads processed using the Iso-Seq bioinformatics pipeline (see Methods)[9]. We detected an average of 10,950 (SD = 615) genes and 38,930 (SD = 2016) transcripts with a mean length of 2.7 kb (SD = 1.6 kb) and 9.88 exons (SD = 7.56 kb) across all samples. There were no differences in gene or transcript

characteristics between genotype groups (WT vs TG mice) or age groups (2 vs 8 months) (Supplementary Fig. 1, Supplementary Data 2). As expected, human-specific *MAPT* transcripts were only detected in TG mice, confirming the stable activation of the human *MAPT* transgene (Supplementary Fig. 2). Raw sequencing data and annotated transcripts from both WT and TG mice are available as a downloadable resource (see Data Availability).

Across all differentially expressed genes (DEGs) previously associated with the progression of tau pathology in TG mice[5], we found a strong overall correlation of effect sizes (corr = 0.60, $P = 5.17 \times 10^{-224}$) between short-read RNA-Seq and long-read Iso-Seq reads, with a significant enrichment of consistent direction of effects (binomial test: $P = 1.45 \times 10^{-15}$, Supplementary Fig. 3). Our short-read RNA-Seq analysis had previously highlighted *Gfap*, encoding glial fibrillary acidic protein (GFAP), as the top-ranked DEG associated with the progression of tau pathology in TG mice[5]. *Gfap* is highly expressed in astrocytes and upregulated in reactive astrocytes in response to brain pathology and neurodegenerative disease[13]. Gene-level analysis of our Iso-Seq reads confirmed this association using both normalized full-length Iso-Seq read counts as a proxy for transcript abundance ($\log_2$ fold change ($\log$2FC) = 3.37, FDR = $1.99 \times 10^{-4}$, Fig. 2A, Supplementary Data 3) and a 'hybrid' approach involving the alignment of short-read RNA-Seq reads from the full set of samples ($n = 30$ WT, 29 TG at ages 2, 4, 6 and 8 months) to the transcript annotations derived from our Iso-Seq data ($\log$2FC = 2.76, FDR = $2.51 \times 10^{-17}$, Fig. 2B). We leveraged our full-length Iso-Seq reads to identify the specific differentially expressed transcript (DET) of *Gfap* (LR.Gfap.16, Fig. 2C)−the most abundantly expressed *Gfap* isoform (Gfap-201, ENSMUST00000067444.9) in the mouse EC – that was the primary driver of overall differential *Gfap* expression in TG mice ($\log$2FC = 3.57, FDR = $1.16 \times 10^{-3}$, Fig. 2D, Supplementary Data 4). LR.Gfap.16 upregulation was also confirmed using the hybrid approach ($\log$2FC = 1.06, $P = 2.1 \times 10^{-2}$) (Fig. 2E). Of note, the hybrid approach also identified the upregulation of several novel *Gfap*-associated isoforms; given the near perfect-homology of these transcripts to LR.Gfap.16 (Fig. 2C), we suspect this reflects the limited sensitivity of short RNA-Seq reads to differentiate similar isoforms, highlighting a key advantage of using long-read sequencing reads to directly quantify transcript expression.

We used fluorescence-activated nuclei sorting (FANS)[12] followed by ONT nanopore sequencing to profile full-length transcripts in purified NeuN+ (neuron-enriched) and NeuN- (glia-enriched) cortical nuclei populations isolated from a subset of mice ($n = 8$, 2 WT and 2 TG at ages 2 and 8 months, Supplementary Data 1). We confirmed an overall upregulation of Gfap-201 (the reference annotated transcript of LR.Gfap.16) in TG mice compared to WT mice (linear regression: $P = 3.76 \times 10^{-2}$, Fig. 2F), with notably higher expression in TG mice at 8 months than at 2 months. Of note, these data showed that this upregulation was exclusively driven by changes in the glia-enriched nuclei population (linear regression: $P = 2.21 \times 10^{-2}$, Fig. 2F). Aggregated transcript tracks for both neuron- and glia-enriched nuclei populations are available as a resource (see Data Availability).

### Ultradeep targeted long-read cDNA sequencing identifies hundreds of rare novel isoforms

The constraints on sample numbers and sequencing depth associated with the whole transcriptome Iso-Seq approach meant we had relatively limited sensitivity to detect and quantify rare transcripts and identify novel between-group differences using the direct quantification of full-length reads. To enable a more comprehensive AS analysis associated with tau pathology, we therefore performed ultra-deep long-read cDNA sequencing on a targeted panel of 20 genes previously implicated in AD (Table 1) in a larger number of WT and TG mice spanning four time-points ($n = 24$, 3 WT and 3 TG at ages 2, 4, 6 and 8 months, Supplementary Data 1). We targeted genes affected by autosomal dominant mutations associated with neurodegenerative

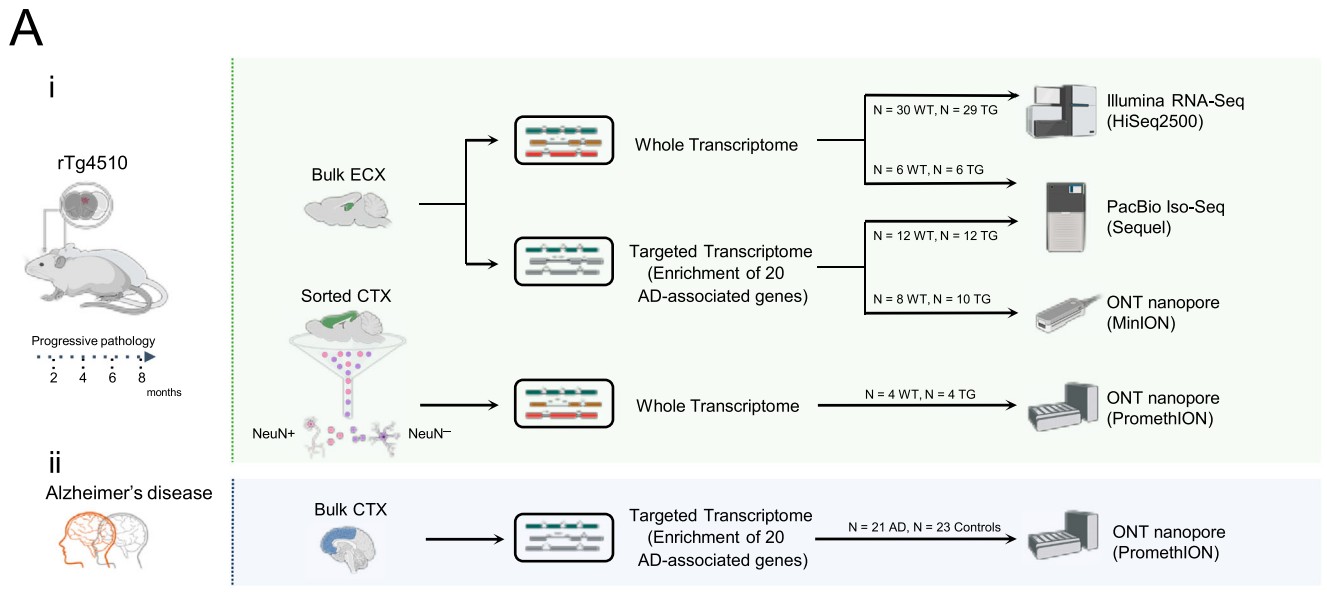

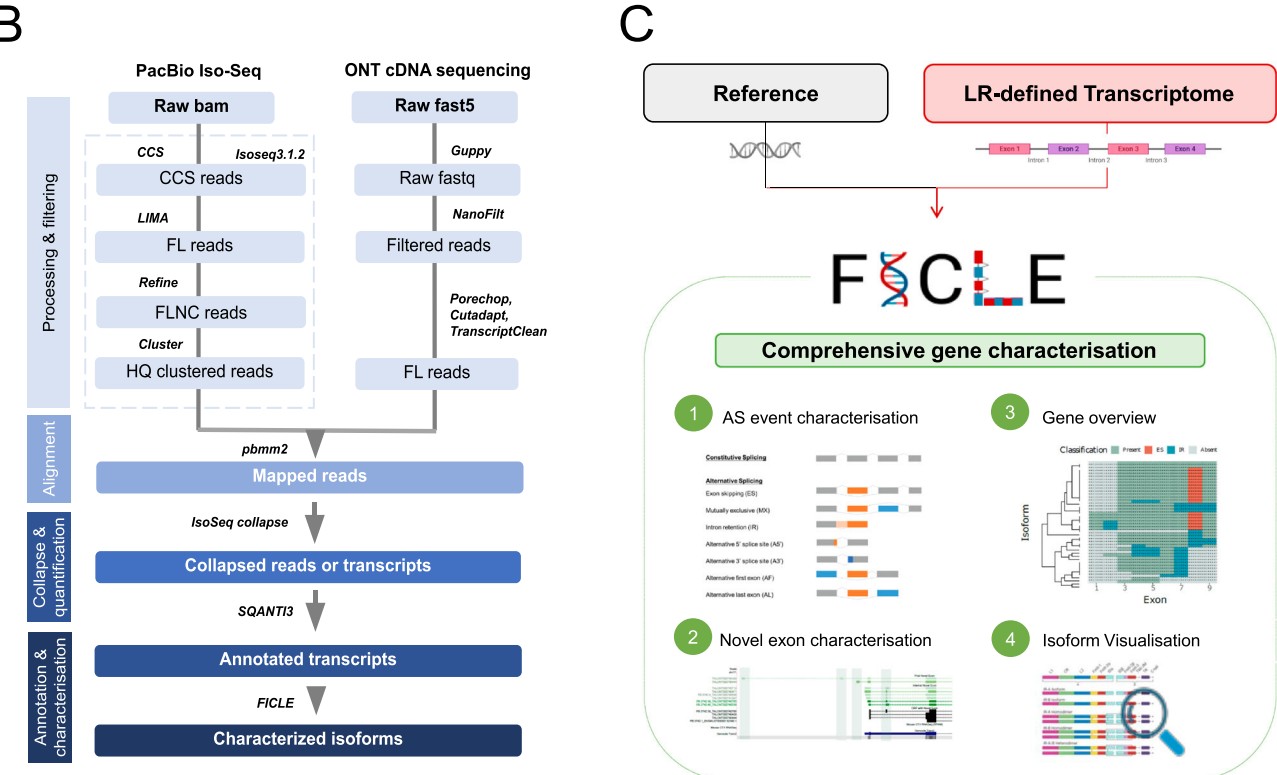

**Fig. 1 | Methodological overview. A** An overview of the experimental approach used to profile isoform diversity in (**i**) the rTg4510 mouse model using short-read RNA-Seq (Illumina), long-read whole transcriptome and targeted Iso-Seq (PacBio) and long-read nanopore cDNA sequencing (ONT) from bulk cortex tissue and FANS-isolated nuclei populations and (**ii**) post-mortem human cortex tissue from individuals with high and low AD neuropathology using long-read targeted nano-pore cDNA sequencing (ONT). Further details on all samples used in this study can be found in Methods, Supplementary Data 1 and Supplementary Data 2. **B** An overview of the bioinformatics pipeline used to process, align, quantify, annotate and characterize full-length transcript reads. Further details are provided in Methods. **C** An overview of the *FICLE* bioinformatics tool developed as part of this work to comprehensively characterize and visualize targeted long-read RNA sequencing data. AD Alzheimer's disease, AS alternative splicing, CCS circular consequence sequence, CTX cortex, ECX entorhinal cortex, FANS fluorescence-activated nuclei sorting, FL full-length, NeuN neuronal nuclear protein, HQ high-quality, LR long-read, ONT Oxford Nanopore Technologies, PacBio Pacific Bios-ciences, WT wild-type mice, TG rTg4510 transgenic mice.

disorders (*App, Mapt, Snca, Fus, Tardbp*), genes identified from both AD genome-wide association studies (GWAS) (*Abca1, Abca7, Apoe, Bin1, Cd33, Clu, Picalm, Ptk2b, Sorl1, and Trem2*) and epigenome-wide association studies (EWAS) (*Ank1, Rhbdf2*), in addition to several genes implicated in related neurodegenerative phenotypes (*Fyn,* *Trpa1, Vgf*). Targeted enrichment was undertaken using custom-designed probes for every known exon across each gene (see Meth-ods), followed by sequencing using both ONT nanopore and PacBio Iso-Seq on EC samples from WT and TG mice (see Fig. 1A and Sup-plementary Data 1).

**Table 1 | Diversity of transcripts expressed from the 20 genes characterized using ultra-deep targeted long-read sequencing (ONT nanopore and PacBio Iso-Seq)**

| Target genes | Number of isoforms (merged ONT and PacBio dataset) | | | | | Number of isoforms with AS events (total number of AS events) | | |
| --- | --- | --- | --- | --- | --- | --- | --- | --- |
| | Classification | | | Coding potential | | | | |
| | All | Known | Novel | Coding | Non-coding | A5A3 | ES | IR |
| Abca1[49] | 44 | 34 | 10 | 43 | 1 | 30 (45) | 3 (3) | 0 (0) |
| Abca7[50] | 82 | 36 | 46 | 81 | 1 | 56 (80) | 26 (47) | 1 (1) |
| Ank1[51] | 54 | 44 | 10 | 44 | 10 | 31 (40) | 37 (116) | 0 (0) |
| Apoe | 1408 | 13 | 1395 | 1348 | 60 | 1380 (2650) | 1303 (4316) | 11 (11) |
| App[52] | 1119 | 147 | 972 | 1074 | 45 | 909 (1436) | 1037 (5889) | 0 (0) |
| Bin1[26] | 624 | 52 | 572 | 605 | 19 | 476 (700) | 571 (2821) | 0 (0) |
| Cd33[53] | 54 | 16 | 38 | 51 | 3 | 50 (91) | 52 (83) | 9 (10) |
| Clu[26] | 964 | 28 | 936 | 937 | 27 | 915 (1758) | 705 (3894) | 1 (1) |
| Fus[26] | 312 | 74 | 238 | 282 | 30 | 234 (292) | 281 (1428) | 36 (36) |
| Fyn[54] | 96 | 45 | 51 | 93 | 3 | 72 (98) | 79 (286) | 0 (0) |
| Mapt[55] | 111 | 22 | 89 | 72 | 39 | 94 (132) | 88 (404) | 0 (0) |
| Picalm[56] | 323 | 140 | 183 | 232 | 91 | 176 (206) | 310 (839) | 0 (0) |
| Ptk2b[57] | 672 | 85 | 587 | 667 | 5 | 513 (811) | 416 (1287) | 1 (1) |
| Rhbdf2[58] | 18 | 12 | 6 | 18 | 0 | 9 (10) | 2 (2) | 0 (0) |
| Snca[59] | 725 | 14 | 711 | 281 | 444 | 690 (1382) | 712 (1732) | 1 (1) |
| Sorl1[60] | 213 | 137 | 76 | 199 | 14 | 197 (280) | 94 (94) | 0 (0) |
| Tardbp[61] | 191 | 79 | 112 | 150 | 41 | 113 (157) | 188 (907) | 69 (69) |
| Trem2[62] | 97 | 13 | 84 | 80 | 17 | 86 (152) | 90 (129) | 5 (5) |
| Trpa1[63] | 4 | 4 | 0 | 4 | 0 | 1 (1) | 0 (0) | 0 (0) |
| Vgf[64] | 51 | 5 | 46 | 44 | 7 | 50 (75) | 37 (38) | 0 (0) |

Shown for each gene are the number of known and novel isoforms, further classified by coding potential, the number of isoforms characterized by alternative splicing events (*A5A3* alternative 5′ and 3′ splice site, *ES* exon skipping, *IR* intron retention) and the total number of these events across all associated isoforms.

Our first aim was to characterize the transcriptional landscape for each of the targeted genes in the mouse EC. Raw reads from ONT and Iso-Seq experiments were merged (Fig. 1B, see Methods) and following stringent quality control, our combined dataset comprised of 5.06 million reads annotated to the 20 genes, with no significant difference in read depth between WT and TG mice (mean number of reads per sample: ONT: WT = 248,117 reads, TG = 277,560 reads, Mann-Whitney-Wilcoxon test: $W = 50$, $P = 0.399$; Iso-Seq: WT = 12,235 reads, TG = 12,424 reads, Mann-Whitney-Wilcoxon test: $W = 81$, $P = 0.624$). As expected, targeted sequencing yielded significantly more unique transcripts of the targeted genes than were detected using whole transcriptome sequencing when comparing data from the 12 samples processed using both approaches (unique transcripts in targeted = 815, unique transcripts in whole transcriptome = 15, common transcripts in both datasets = 440; fold-change = 2.76, Fisher's Exact test: $P = 4.07 \times 10^{-134}$) (Fig. 3A). We again confirmed expression of human-specific *MAPT* transcripts only from TG mice in the targeted datasets (Supplementary Fig. 2).

The depth of sequencing achieved using our enrichment approach enabled us to detect 81,221 unique transcripts annotated to the 20 target genes. The vast majority of these transcripts were both novel ($n = 77,837$ (95.8%)) and very rare ($n = 74,411$ (91.6%) of detected transcripts were characterized by <10 full-length reads across any five samples) (Supplementary Fig. 4). We detected an average of ~2200 rare transcripts for each target gene that individually comprised <1% of sequencing reads, but collectively constituted a relatively large proportion of total reads. Of note, ~50% of the on-target reads were attributed to five transcripts representing known isoforms of *Apoe*, *Clu* and *Snca* (Fig. 3B). Virtually all transcripts identified in the PacBio Iso-Seq dataset ($n = 2060$ transcripts (92%)) were also detected by ONT sequencing, whereas a smaller proportion of transcripts identified in the ONT dataset were also detected in the PacBio Iso-Seq dataset, reflecting the much higher number of reads obtained from ONT

sequencing (Fig. 3C). Despite the large number of novel transcripts identified, most target genes were characterized by a relatively small number of 'major' isoforms (mean = 9.4 isoforms, SD = 10.6, Fig. 3D), with a transcript defined as 'major' if its proportion relative to the dominant (most abundant) isoform was >0.5 (see Methods). This corroborates data from existing datasets such as VastDB – the largest resource of AS events in vertebrates – which shows that the majority of genes were characterized by the simultaneous expression of a relatively small number of major isoforms[14].

## FICLE—a novel tool for isoform characterization from long-read sequencing data

Existing bioinformatic tools[15,16] for characterizing AS events using short-read RNA-Seq data fail to capture the connectivity and complexity of transcripts detected from ultra-deep long-read sequencing. Consequently, to comprehensively annotate the full repertoire of transcripts in our data, we developed a novel bioinformatics tool – *FICLE* (see Methods and Fig. 1C) – which is available as a resource to the community (see Code Availability). *FICLE* directly compares the splice junctions between long-read-derived and reference transcripts, enabling a gene-level overview of all detected isoforms and splicing patterns. It accurately characterizes multiple AS events for each transcript, identifies novel 'cryptic' exons via the inclusion of intronic sequences in a transcript, and enables the visualization of isoforms classified by AS events. *FICLE* can be implemented to provide detailed isoform characterization and aid biological interpretations, with the generation of multiple output summary tables and graphs at the exon and transcript level.

## Widespread isoform diversity across genes implicated in Alzheimer's disease

After stringent filtering of rare transcripts by removing transcripts with <10 full-length reads across all samples (Supplementary Fig. 4), our

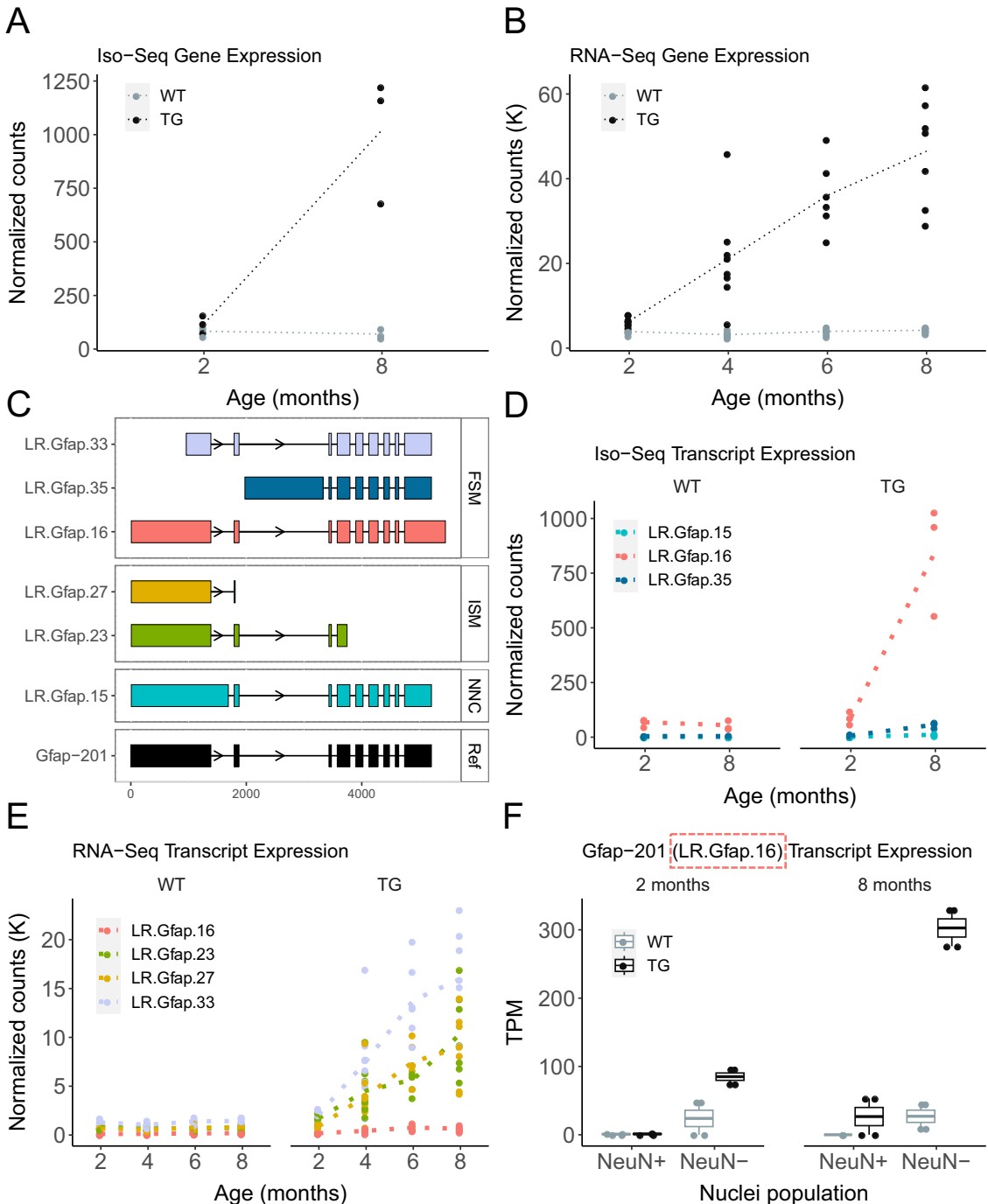

**Fig. 2 | The upregulation of specific *Gfap* transcripts is associated with the progression of tau pathology in TG mice. A** *Gfap* expression determined from normalized PacBio Iso-Seq full-length read counts and **B** normalized short-read RNA-Seq read counts in WT (gray) and TG (black) mice (Iso-Seq: *n* = 12, RNA-Seq: *n* = 59). **C** Visualization of the top-ranked differentially expressed *Gfap* transcripts associated with the progression of tau pathology. *Gfap* transcript expression is determined from (**D**) full-length Iso-Seq reads and (**E**) RNA-Seq reads aligned to long-read annotations (hybrid approach). Colors refer to transcripts shown in **C. F** Box-plot of normalized read counts (TPM) of the Gfap-201 transcript

(corresponding to the LR.Gfap.16 transcript in the targeted dataset) detected in NeuN+ (neuron-enriched) and NeuN- (glia-enriched) nuclei populations from WT and TG mice at 2 and 8 months (*n* = 8). Dotted lines represent the mean paths across age (months). The middle box represents the interquartile range (IQR), the middle line represents the median, and the whisker lines represent the minimum (quartile 1 – 1.5 × IQR) and the maximum (quartile 3 + 1.5 × IQR). K thousand, LR long-read, NeuN neuronal nuclear protein, WT wild-type mice, TG rTg4510 transgenic mice, TPM transcripts per million.

final dataset included 7162 isoforms representing 1000 (14%) known isoforms and 6162 (86%) novel isoforms. The vast majority of the splice junctions in multi-exonic isoforms (*n* = 7158 (99.9%)) were supported by short-read RNA-Seq data generated on the same individuals (*n* = 52,225 splice junctions (90.0%)). Furthermore, the majority of isoforms contained known polyA motifs (*n* = 5587 (78.0%)) and ~40%

were located proximal (within 50 bp) to Cap Analysis Gene Expression (CAGE) peaks from the mouse FANTOM5 dataset[17]. The median number of filtered isoforms across each of the 20 genes was 151, although there was considerable heterogeneity in isoform number between genes. *Apoe* was the most isomorphic gene (1408 isoforms detected of which 13 were known and 1395 were novel) while *Trpa1* was the least

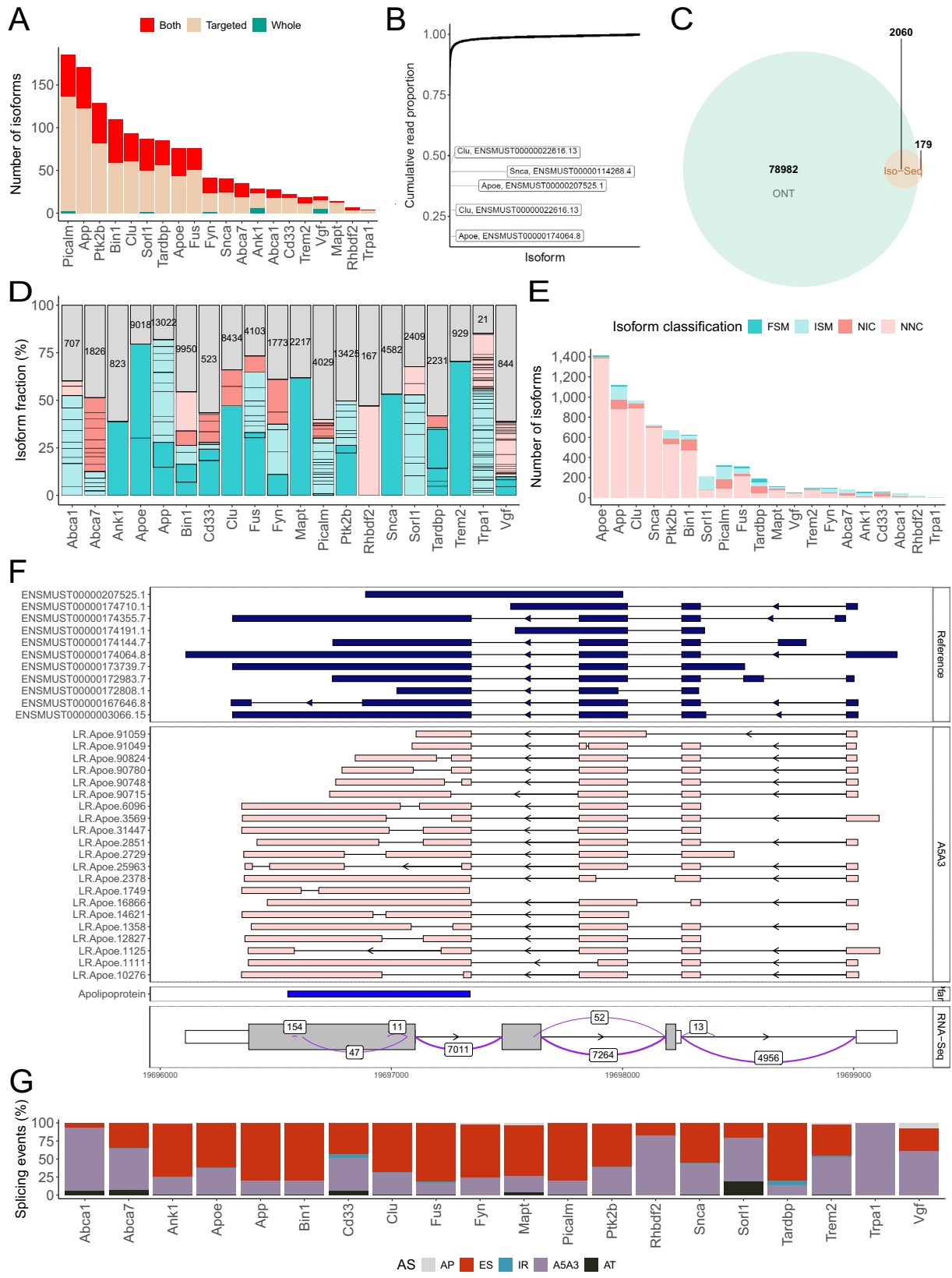

isomorphic gene (4 isoforms detected, all previously characterized) (Fig. 3E). We used *FICLE* to classify *Apoe* isoforms by exonic structure, observing a complex pattern of AS events involving the last exon, which encodes the apolipoprotein domain that binds to lipids, and the 3′ UTR (Fig. 3F). Supported by short-read RNA-Seq data from an overlapping set of samples (Fig. 3F), the variation in the last exon included usage of alternative 5′ and 3′ splice sites and intra-exonic internal skipping, a phenomenon seen in one of the known isoforms (Apoe-202, ENSMUST00000167646.8) (Fig. 3F). Of note, despite being highly isomorphic, *Apoe* is the shortest gene in our targeted gene panel (gene length = 3.08 kb) with the fewest number of exons (*n* = 5) (Supplementary Fig. 5).

**Fig. 3 | Targeted long-read sequencing identifies considerable isoform diversity and alternative splicing events amongst genes implicated in AD pathology.** **A** Isoforms detected in the whole transcriptome and targeted PacBio Iso-Seq datasets generated from an overlapping subset of mouse samples ($n = 12$) across the 20 targeted genes. Commonly detected isoforms using both approaches are depicted in red, and those unique to the targeted and whole transcriptome datasets are depicted in beige and green, respectively. **B** Cumulative proportion of ONT full-length reads in the targeted dataset. Each dot represents a transcript, and the respective proportion is derived using the summation of reads across all samples ($n = 18$). **C** Venn diagram of the total number of transcripts from the 20 AD genes detected using targeted ONT (green) and PacBio Iso-Seq (orange) sequencing. **D** The repertoire of isoforms, and their usage, expressed from the 20 targeted genes, coloured by *SQANTI3* structural category (blue: FSM full splice match, light blue: ISM incomplete splice match, pink: NIC novel in catalog, light pink: NNC novel not in catalog). Gray bars depict 'minor' isoforms. The lines across the bars

delineate the individual isoforms, and the numbers in the gray bars refer to the total number of minor isoforms. The isoform fraction per gene is determined by dividing the mean normalized read counts of the associated isoforms across all the samples by the total normalized mean read counts of all the associated isoforms. **E** The total number of isoforms annotated to each of the 20 targeted genes, colored by *SQANTI3* structural category (FSM, ISM, NIC, NNC) after merging targeted datasets and filtering by expression (see Methods). **F** Visualisation of *Apoe* isoforms detected in the merged targeted datasets, highlighting the complexity of the last exon and 3' UTR, as supported by splice junction coverage from short-read RNA-Seq data generated on an extended set of samples ($n = 30$ WT, 29 TG). **G** The proportion of alternative splicing events (AP alternative promoter, ES exon skipping, IR intron retention, A5A3 alternative 5' and 3' splice sites, AT alternative termination) observed in isoforms annotated to the 20 targeted genes (Table 1) identified using *FICLE*.

Using the sequence of the full-length transcripts identified in the final dataset, we next created a predicted protein database by calling open reading frames (ORFs) and applying an established long-read proteogenomics pipeline[18] (see Methods). In total, 5596 (78.1%) of transcripts were predicted to be protein-coding and incorporated a stop codon. Of these, ~70% were predicted to have a distinct ORF resulting in a total of 3841 unique protein isoforms from the 20 genes (Supplementary Fig. 6). As expected, the number of unique predicted protein isoforms translated from a gene was positively correlated with the number of detected transcripts (Pearson's correlation = 0.95, $P = 1.87 \times 10^{-10}$), and the vast majority ($n = 1248$ (71.1%)) of 'redundant' transcripts that collapsed into a non-unique ORF were classified as Novel Not in Catalog (NNC) with at least one novel splice site (Supplementary Fig. 6). The majority of the predicted protein isoforms were classified as protein Novel Not in Catalog (pNNC) ($n = 3044$, 79.2%), derived from NNC transcripts and containing at least one novel element (i.e. a novel N-terminus or splice junction) (Supplementary Data 5).

**Extensive alternative splicing events across targeted AD genes**
Using *FICLE*, we identified widespread AS events associated with each of the 20 targeted genes (Table 1 and Fig. 3G). In line with our previous findings[9], we observed extensive use of alternative 5' (A5') and 3' splice site (A3') ($n = 10,396$ events observed in 6082 (84.9%) isoforms) and exon skipping (ES) ($n = 24,315$ events observed in 6031 (84.2%) isoforms) (Supplementary Fig. 7). Although the majority of isoforms were characterized by the skipping of several exons, some genes expressed transcripts that were characterized by highly elevated levels of ES. For example, over 10% of isoforms expressed from *App* ($n = 115$ isoforms (10.3%)) and *Bin1* ($n = 72$ isoforms (11.5%)) were characterized by the skipping of > 10 exons (Supplementary Fig. 7). In contrast, intron retention (IR) events were less frequent ($n = 135$ events observed in 134 isoforms (1.87%)) and only detected in isoforms expressed from a subset of the target genes (*Abca7, Apoe, Cd33, Clu, Fus, Ptk2b, Snca, Tardbp* and *Trem2*), corroborating our previous findings[9,19]. Only one isoform (a novel *Cd33* isoform) was characterized with more than one IR event (Supplementary Fig. 8), although the majority of IR events ($n = 126$ events (93.3%)) spanned two or more exons with *Tardbp*-associated transcripts demonstrating extensive IR spanning up to 11 exons (Supplementary Fig. 8). Finally we detected transcripts incorporating novel cryptic exons (CE) annotated to *Apoe* ($n = 1$), *Bin1* ($n = 7$), *Clu* ($n = 1$), *Snca* ($n = 1$) and *Trem2* ($n = 7$) (Supplementary Fig. 9). The CE-containing transcripts annotated to *Apoe, Clu, Bin1* and *Trem2* were in-frame with no premature stop codons, and either escaped nonsense-mediated decay (NMD) or produced shorter 'non-coding' ORFs. In contrast, we observed an out-of-frame CE in a novel *Snca* transcript (Supplementary Fig. 9) located within the synuclein domain;

this transcript incorporates a premature termination codon and is predicted to undergo NMD.

**Differential transcript expression associated with the progression of tau pathology in TG mice**
We used our ultradeep ONT sequencing data to identify specific transcripts associated with the progression of tau pathology in TG mice (see Methods), identifying 12 differentially expressed transcripts (DETs) expressed from six of the 20 target genes (*Apoe, App, Cd33, Clu, Fyn,* and *Trem2*) at a FDR < 0.05 (Fig. 4A, Table 2, Supplementary Data 6). Notably, these transcripts were all up-regulated in TG mice with the progression of tau pathology and five (annotated to *Clu, Fyn,* and *Apoe*) represented novel transcripts not present in existing transcript annotations (Fig. 4A). All 12 of the differentially expressed transcripts were also detected in our PacBio Iso-Seq replication dataset and differentially expressed with the same direction of effect and a strong overall concordance in effect sizes between the two datasets (Log2FC pathology: Spearman's correlation = 0.643, $P = 0.03$). Three DETs were confirmed at a Bonferroni-corrected $P$-value ($P < 4.2 \times 10^{-3}$) − a known *Trem2* isoform (LR.Trem2.54), a novel *Clu* (LR.Clu.39341) and novel *App* (LR.App.7564) isoform − and eight additional DETs reached nominal significance ($P < 0.05$) despite the considerably lower depth of the PacBio sequencing and reduced power for differential expression analyses (Supplementary Data 6).

We identified an additional 48 DETs (FDR < 0.05) expressed from 10 genes (*Abca7, App, Bin1, Cd33, Mapt, Picalm, Ptk2b, Rhbdf2, Snca, Trem2*) associated with genotype (i.e characterized by an overall significant difference in expression between WT and TG mice) (Fig. 4B, Supplementary Data 7 and Supplementary Fig. 10). The majority of these DETs ($n = 28$) were up-regulated in TG mice and a large proportion ($n = 14$) were annotated to *Trem2* (see below). Of note, the top-ranked up-regulated transcripts in TG mice were isoforms of *Mapt* (Fig. 4); interestingly, these did not correspond to transcripts derived from the human transgene suggesting the transgene induces concomitant effects on endogenous *Mapt* transcript expression. Among the DETs ($n = 31$) also detected in the PacBio Iso-Seq dataset, all were characterized by a consistent direction of effect and there was a strong overall concordance in effect sizes with the ONT data (Log2FC genotype: Spearman's correlation = 0.792, $P = 3.78 \times 10^{-7}$). Finally, we explored differences in *relative* isoform abundance (isoform fraction (IF)) associated with tau pathology, testing for the switching of the dominant (i.e. most highly expressed) isoform between experimental groups using *EdgeR spliceVariants* (see Methods). We identified three genes (*Fus, Bin1,* and *Trpa1*) characterized by significant differential transcript usage (DTU) between WT and TG mice, and three genes (*Bin1, Vgf,* and *Clu*) characterized by DTU associated with the progression of tau pathology in TG mice (Supplementary Fig. 11 and Supplementary Data 8).

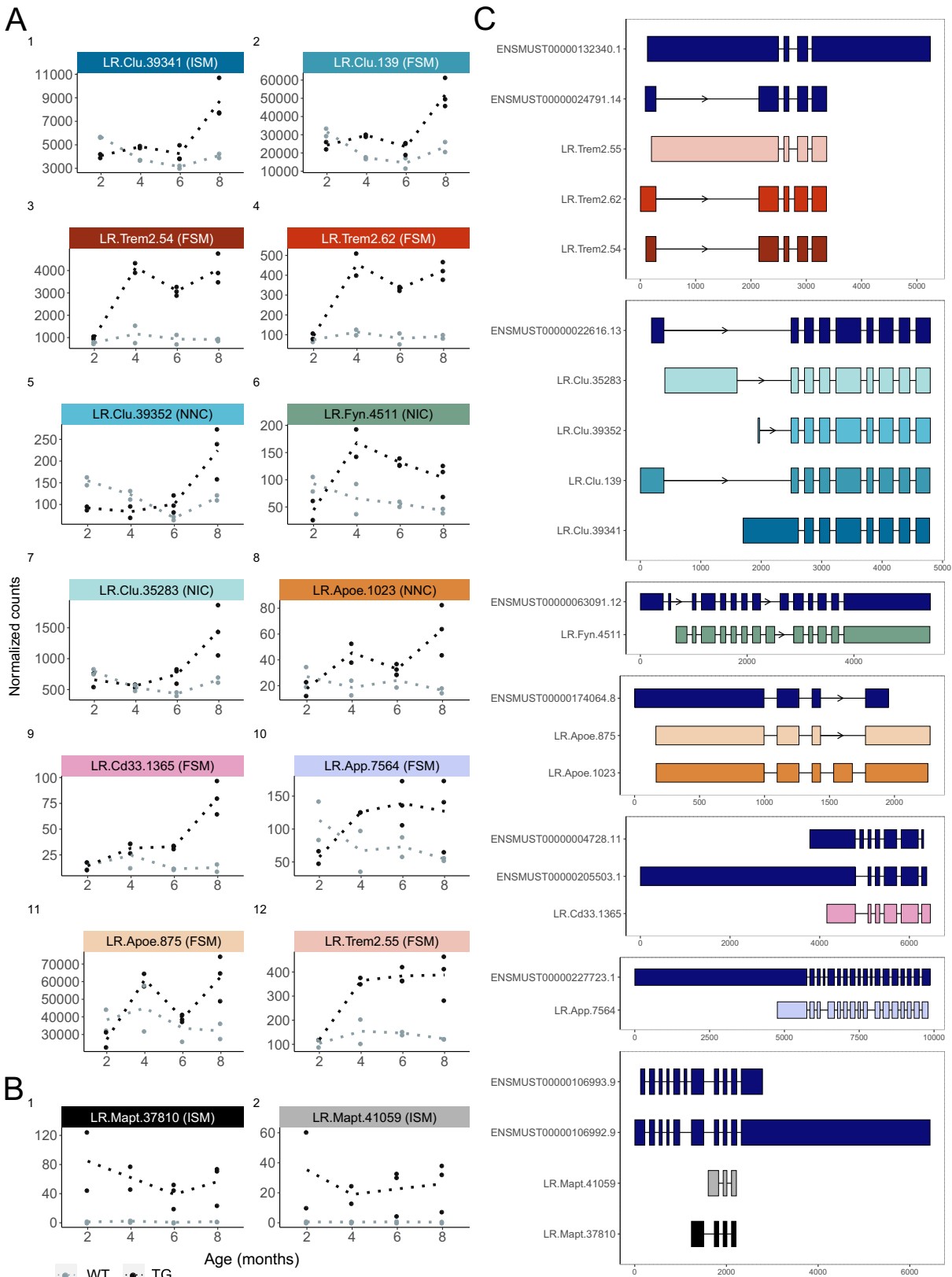

**Fig. 4 | Differential transcript expression associated with the progression of tau pathology in TG mice. A** Transcript expression (ONT full-length read counts) of the 12 significantly upregulated transcripts in TG mice (black) associated with the progression of tau pathology and **B** the two top-ranked endogenous *Mapt* transcripts significantly up-regulated in TG compared to WT mice. Plots in **A** and **B** are ordered by significance. **C** Structure of these 14 transcripts alongside reference transcripts for each gene. Transcripts are color-coded by gene, with shading indicating significance (darker indicates greater significance). *SQANTI3* structural categories (FSM full splice match, ISM incomplete splice match, NIC novel in catalog, NNC novel not in catalog) are provided in parenthesis for each transcript.

**Table 2 | Significant differentially expressed transcripts associated with tau pathology in TG mice**

| Isoform | Gene | Associated Transcript (Structural category) | Discovery (ONT) | | | | Replication (Iso-Seq) |
|---|---|---|---|---|---|---|---|
| | | | log2FC | lfcSE | *P*-value | FDR | *P*-value |
| LR.Clu.39341 | *Clu* | ENSMUST00000022616.13 (ISM) | 1.57 | 0.33 | 1.72E−06 | 5.72E−04 | 4.89E−03 |
| LR.Clu.139 | *Clu* | ENSMUST00000022616.13 (FSM) | 1.53 | 0.37 | 2.64E−05 | 4.06E−03 | 6.80E−01 |
| LR.Trem2.54 | *Trem2* | ENSMUST00000024791.14 (FSM) | 1.75 | 0.43 | 4.16E−05 | 4.06E−03 | 5.12E−05 |
| LR.Trem2.62 | *Trem2* | ENSMUST00000113237.3 (FSM) | 1.88 | 0.46 | 4.88E−05 | 4.06E−03 | 2.57E−02 |
| LR.Clu.39352 | *Clu* | Novel (NNC) | 1.69 | 0.47 | 2.87E−04 | 1.60E−02 | 2.22E−01 |
| LR.Fyn.4511 | *Fyn* | Novel (NIC) | 2.26 | 0.62 | 2.58E−04 | 1.60E−02 | 7.35E−03 |
| LR.Clu.35283 | *Clu* | Novel (NIC) | 1.42 | 0.41 | 4.64E−04 | 2.21E−02 | 8.47E−04 |
| LR.Apoe.1023 | *Apoe* | Novel (NNC) | 2.51 | 0.74 | 7.30E−04 | 2.98E−02 | 4.88E−01 |
| LR.Cd33.1365 | *Cd33* | ENSMUST00000205503.1 (FSM) | 2.66 | 0.79 | 8.04E−04 | 2.98E−02 | 8.70E−01 |
| LR.App.7564 | *App* | ENSMUST00000227021.1 (FSM) | 2.18 | 0.66 | 1.03E−03 | 3.12E−02 | 4.07E−04 |
| LR.Apoe.875 | *Apoe* | ENSMUST00000174064.8 (FSM) | 1.47 | 0.45 | 9.64E−04 | 3.12E−02 | 3.87E−02 |
| LR.Trem2.55 | *Trem2* | ENSMUST00000132340.1 (FSM) | 1.46 | 0.47 | 1.73E−03 | 4.81E−02 | 7.89E−03 |

Shown are the 12 differentially expressed transcripts associated with tau pathology in TG mice (Fig. 4) using the Wald test in *DESeq2* with ONT full-length read counts as a proxy of transcript expression. Results derived from Iso-Seq full-length read counts (targeted PacBio Iso-Seq dataset) are also shown, and the structural categories from *SQANTI3* are also provided.
*FSM* Full Splice Match, *ISM* Incomplete Splice Match, *NIC* Novel in Catalog, *NNC* Novel Not in Catalog.

## Increased abundance of intron retention, nonsense-mediated decay and cryptic exon inclusion in TG mice

Recent studies have identified an increase in transcripts with intron retention (IR)[20] and deficits in NMD activity[21] associated with AD. Although there was no difference in the abundance of specific transcripts characterized with IR and NMD between TG and WT mice, there was a significant increase in the total abundance of *Trem2* and *Cd33* transcripts characterized with IR (*Trem2*: log2FC = 1.92, $P = 5.05 \times 10^{-4}$; *Cd33*: log2FC = 1.19, $P = 1.66 \times 10^{-2}$, Supplementary Fig. 12) and NMD (*Trem2*: log2FC = 1.58, $P = 4.81 \times 10^{-5}$; *Cd33*: log2FC = 1.16, $P = 5.08 \times 10^{-3}$, Supplementary Fig. 13) in TG mice. We also identified an overall increase in the total abundance of CE-containing transcripts detected in TG compared to WT mice (log2FC = 0.96, $P = 1.77 \times 10^{-3}$), and associated with progression of tau pathology in TG mice ($P = 4.43 \times 10^{-2}$) (Supplementary Fig. 9). These findings align with previous studies demonstrating aberrant splicing events in AD, and suggest that increased intron retention, cryptic exon inclusions and deficits in NMD in transcripts of AD-associated genes are associated with tau pathology.

## Upregulation of the dominant Trem2 transcript is associated with elevated tau pathology in TG mice

After *Mapt*, the top two DETs in TG mice were two known *Trem2* isoforms – LR.Trem2.62 (Trem2-202, ENSMUST00000113237.3) (log2FC = 1.87, FDR = $7.46 \times 10^{-7}$) and LR.Trem2.54 (Trem2-201, ENSMUST00000024791.14) (log2FC = 1.73, FDR = $8.59 \times 10^{-7}$). Both isoforms were also strongly associated with the progression of tau pathology in TG mice (LR.Trem2.62: log2FC = 1.88, FDR = $4.06 \times 10^{-3}$; LR.Trem2.54: log2FC = 1.75, FDR = $4.06 \times 10^{-3}$) (Fig. 4 and Table 2). This association was confirmed in our PacBio Iso-Seq targeted dataset with both transcripts associated with genotype (LR.Trem2.62: log2FC = 1.73, FDR = $5.2 \times 10^{-4}$; LR.Trem2.54: log2FC = 1.55, FDR = $3.3 \times 10^{-6}$) and pathology in TG mice (LR.Trem2.62: log2FC = 2.43, FDR = $9.96 \times 10^{-1}$; LR.Trem2:54: log2FC = 2.20, FDR = $3.85 \times 10^{-2}$). Hierarchical clustering of individual samples based on the expression of detected *Trem2* transcripts (Fig. 5A) confirmed the relationship with tau pathology in aging TG mice (Fig. 5B, C). We used our cell-type-specific transcript annotations to show that both isoforms are primarily expressed in non-neuronal (glia-enriched) nuclei (Trem2-201/LR.Trem2.54: $P = 2.73 \times 10^{-3}$; Trem2-202/LR.Trem2.62: $P = 4.98 \times 10^{-2}$), confirming significantly higher expression in TG compared to WT mice (Trem2-201/LR.Trem2.54: log2FC = 1.79, $P = 2.90 \times 10^{-2}$; Trem2-202/LR.Trem2.62:

log2FC = 6.62, $P = 5.91 \times 10^{-2}$) (Fig. 5D). Immunohistochemistry staining (see Methods) confirmed previous data showing that upregulated *Trem2* in aged TG mice is localized to activated microglia in close contact with phosphorylated tau (Fig. 5E and Supplementary Fig. 14). Of note, a significant increase of these two *Trem2* transcripts was also reported in a recent study using qPCR in two TG amyloid models of AD[22], suggesting that these transcripts are upregulated in response to both tau and amyloid pathology. Notably, the up-regulated mouse LR.Trem2.54 isoform shows high homology with the most abundantly expressed human *TREM2* isoform (ENST00000373113.8, 75.3% homology, 81% query cover)[23].

Overall, *Trem2* is a high isomorphic gene ($n = 97$ transcripts) with 84 detected novel transcripts encoding 50 novel predicted protein isoforms (Supplementary Fig. 6). The expression of one of the tau-associated DETs (LR.Trem2.54) was considerably higher than any of these relatively rare novel transcripts (Fig. 5F) with the upregulation of this isoform being the primary driver of the gene-level increase in *Trem2* gene expression observed in older TG mice (log2FC = 2.20, FDR = $3.84 \times 10^{-2}$) (Fig. 5G). The vast majority of the rare novel *Trem2* transcripts primarily differ in the use of alternative 5' and 3' splice sites ($n = 86$ isoforms (88.7%)) (Fig. 5A, C), particularly in exon 2, and this was confirmed by short-read RNA-Seq data from matched samples (Supplementary Fig. 15). This exon was characterized by relatively few ES events ($n = 6$ isoforms (6.2%)) (Fig. 5A), potentially reflecting its functional importance; it encodes the Ig-like V-type domain that is crucial for ligand interactions (Fig. 5C). Of note, the majority of genetic variants in *TREM2* associated with AD are located in the homologous region of human transcripts[24]. Despite the dramatic upregulation of the dominant *Trem2* isoform in TG mice, we observed no overall difference in relative *Trem2* transcript usage between TG and WT mice (Supplementary Data 8) with transcript proportions remaining stable between groups (Fig. 5H) and across age (Fig. 5I).

## Upregulation of Clu-201 results in differential gene expression and isoform usage associated with tau pathology

Several DETs associated with the progression of tau pathology in TG mice were annotated to *Clu* (Fig. 4, Table 2). The human *CLU* gene encodes a multifunctional glycoprotein that acts as an extracellular chaperone involved in immune regulation and lipid homeostasis[25] and has been strongly implicated in AD by GWAS[26]. We identified considerable complexity in *Clu* transcripts, detecting 964 isoforms (Fig. 6A, B). In particular, there was widespread alternative first exon

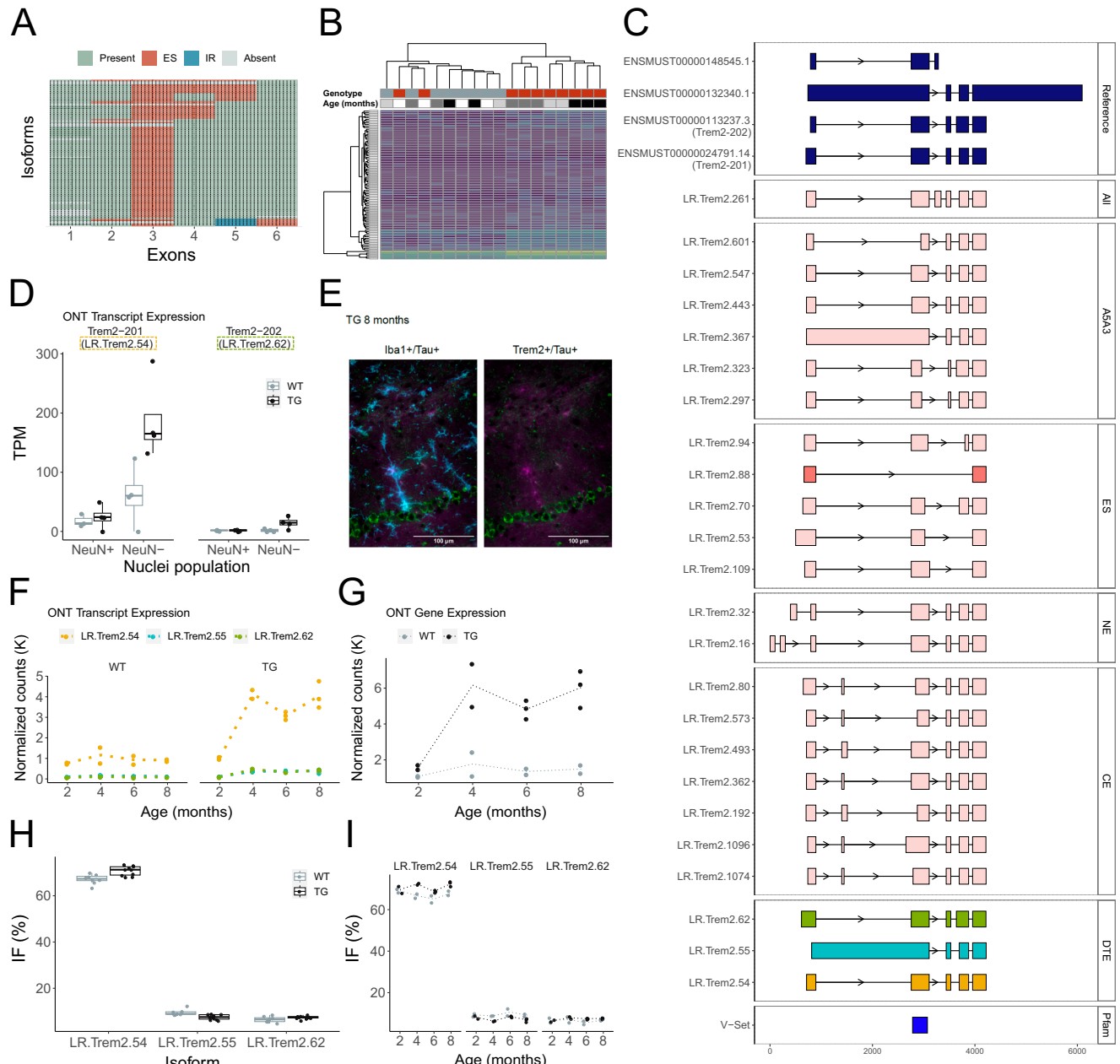

**Fig. 5 | Different isoforms of *Trem2* are up-regulated with the progression of tau pathology in TG mice. A** Cluster dendrogram depicting the repertoire of *Trem2* transcripts identified in the rTg4510 cortex. Each row and column corresponds to a distinct isoform and known exon, respectively. Colors refer to the presence of alternative splicing events (ES exon skipping, IR intron retention). **B** Heatmap depicting the expression of all *Trem2* isoforms. Each row refers to an isoform, colored by structural category (FSM – dark blue, ISM – light blue, NIC – pink, NNC – light pink), and each column refers to a sample with the genotype (WT – gray, TG – red) and age (2 months – white, 4 months – light gray, 6 months – dark gray, 8 months – black) shown. **C** Visualization of the reference and selected novel *Trem2* isoforms coloured by structural category (pink – NIC, light pink – NNC), including a novel isoform incorporating all six known exons (All), novel isoforms with alternative 5' and 3' splice sites (A5A3), exon skipping (ES), and novel exons located upstream (NE) and within the gene (CE). Shown in the final

panel are the three differentially expressed transcripts (DETs) associated with tau pathology, with colors matching **F. D** Box-plot of normalized read counts (TPM) of Trem2-201 and Trem2-202 transcripts detected in NeuN+ (neuron-enriched) and NeuN- (glia-enriched) nuclei populations from WT and TG mice (*n* = 8). **E** Representative immunohistochemistry staining of hippocampal tissue from rTg4510 TG mice at 8 months for Iba1 (cyan), Trem2 (red) and AT8Tau (green). Staining was repeated across 4 sections and uncropped scans are provided in Supplementary Fig. 19. **F** Transcript-level expression of three *Trem2* DETs associated with tau pathology. (**G**) *Trem2* gene expression in WT (gray) and TG (black) mice across age (*n* = 18). Isoform usage (proportion) of the three *Trem2* DETs stratified by **H** genotype and **I** age (*n* = 18). For box-plots, the middle box represents the interquartile range (IQR), the middle line represents the median, and the whisker lines represent the minimum (quartile 1 – 1.5 × IQR) and the maximum (quartile 3 + 1.5 × IQR). Expression is derived from normalizing ONT full-length read counts.

usage with six distinct alternative first exons, although the majority of *Clu* transcripts (*n* = 520, 54%) incorporated the first upstream exon of the known dominant isoform (Clu-201, ENSMUST00000022616.13) (Fig. 6A), skipping these alternative first exons and including the full-length of the clusterin domain (Fig. 6B). We also found evidence of

significant ES and IR events localized to other regions of the gene (exon 9 skipping: *n* = 201 isoforms, exon 10 skipping: *n* = 225 isoforms, exon 11 skipping: *n* = 291 isoforms) (Fig. 6A).

One of the top-ranked DETs associated with progressive tau pathology in TG mice was LR.Clu.139 (log2FC = 1.13, FDR = 2.73 × 10⁻⁴)

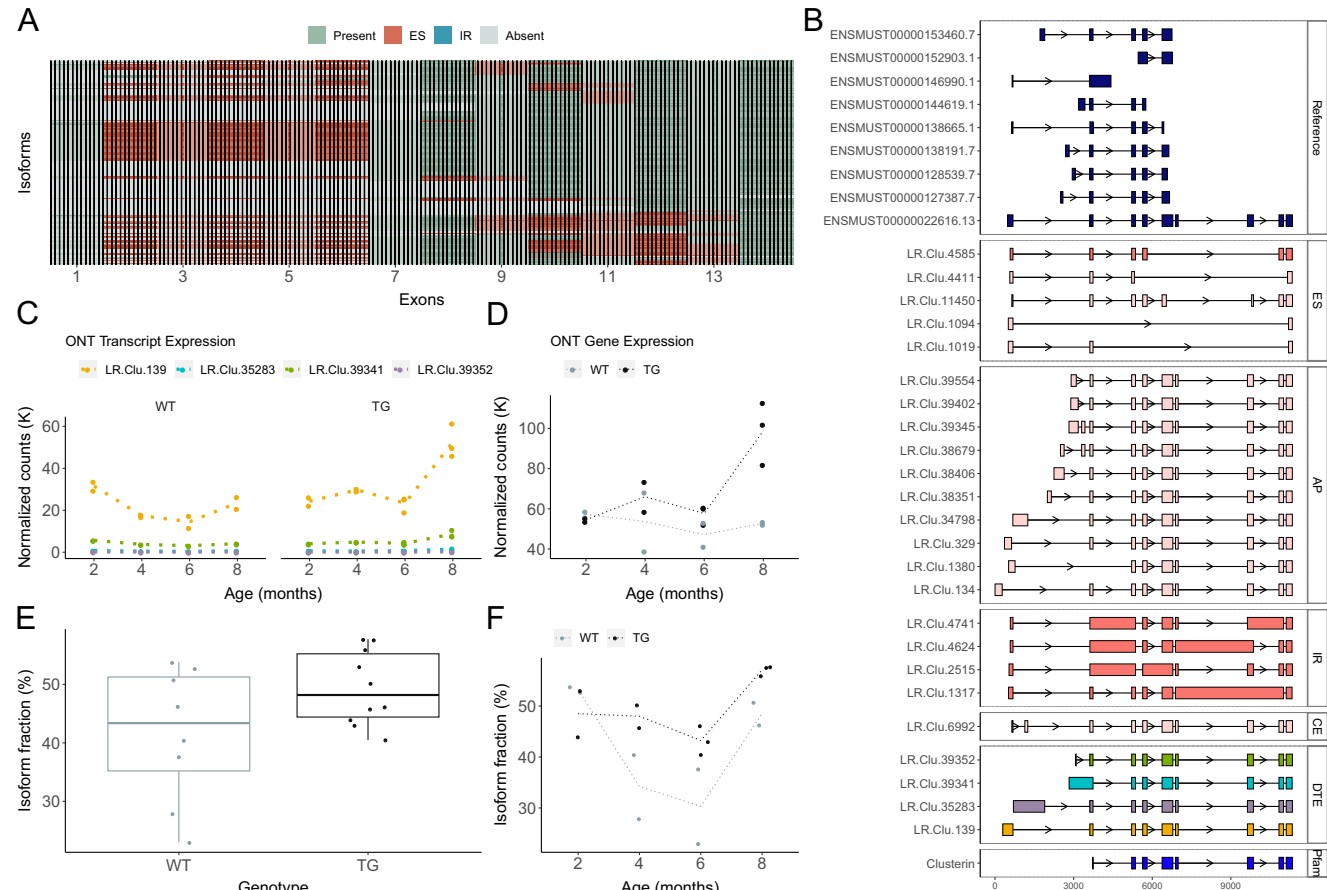

**Fig. 6 | Upregulation of Clu-201 result in differential gene expression and isoform usage. A** A cluster dendrogram depicting the repertoire of *Clu* isoforms detected using targeted long-read sequencing in the rTg4510 cortex. Each row corresponds to a distinct isoform and each column represents a known exon. Colors refer to the presence of alternative splicing events (ES exon skipping, IR intron retention). **B** Visualization of the reference and selected novel isoforms of *Clu* characterized by exon skipping (ES), intron retention (IR), alternative promoters (AP) and novel cryptic exons (CE). Isoforms are coloured by structural category (pink – NIC, light pink – NNC). Shown in the final panel are the four DETs associated with tau pathology, and colors refer to transcripts shown in **C**.

**C** Transcript expression of the four *Clu* DETs associated with tau pathology (see also Fig. 4). **D** *Clu* gene expression (normalized ONT full-length read counts) in WT (gray) and TG (black) mice across age. Isoform usage (proportion) of LR.Clu.139 stratified by (**E**) genotype and (**F**) age (*n* = 18). For the shown box-plot, the middle box represents the interquartile range (IQR), the middle line represents the median, and the whisker lines represent the minimum (quartile 1 – 1.5 × IQR) and the maximum (quartile 3 + 1.5 × IQR). DET differentially expressed transcript, FSM full splice match, ISM incomplete splice match, NIC novel in catalog, NNC novel not in catalog, WT wild-type mice, TG rTg4510 transgenic mice.

(Figs. 4 and 6C), which corresponds to the known dominant *Clu* isoform (Clu-201). Three other *Clu* transcripts that primarily differ from LR.Clu.139 by the inclusion of a novel alternative first exon (Figs. 4B and 6B) were also up-regulated in older TG mice (Fig. 6C). Increased expression of these four *Clu* transcripts was reflected in an overall increase of *Clu* gene expression associated with increased tau pathology in TG mice (log2FC = 0.97, FDR = 1.1 × 10⁻²) (Fig. 6D). Of note, there was a notable shift in isoform usage with the proportion of LR.Clu.139 relative to other *Clu* isoforms (Supplementary Fig. 11) increasing with the progression of pathology in TG mice (total change = 29.2, FDR = 3.12 × 10⁻²) (Fig. 6E, F). These findings corroborate previous studies on human post-mortem tissue reporting increased expression of the two canonical human *CLU* isoforms in AD, which similarly share a common 3' exonic structure (exons 2 – 9) but differ in the use of exon 1 and proximal promoters[25].

### Differential transcript use of Bin1 isoforms associated with the progression of tau pathology

The human Bridging Integrator 1 gene (*BIN1*), which is involved in membrane protein trafficking, is another locus strongly implicated in AD by GWAS[26]. In rTg4510 mice, *Bin1* was another gene characterized by major DTU associated with both genotype (total change = 17.9,

FDR = 2.04 × 10⁻²) and the progression of tau pathology (total change = 30.4, FDR = 7.45 × 10⁻¹⁴). We found *Bin1* to be highly isomorphic in the mouse cortex (*n* = 624 isoforms) with widespread exon skipping that was particularly focused within specific regions of the gene (Fig. 7A, B). As reported for the human *BIN1* gene[27], the inclusion of exons 14 – 16, which encode the conserved CLAP domain involved in endocytosis[28], was highly variable among *Bin1* transcripts in the mouse cortex (exon 14 skipping: *n* = 326 isoforms, exon 15 skipping: *n* = 242 isoforms, exon 16 skipping: *n* = 459 isoforms) (Fig. 7B). In contrast, the first 10 exons of *Bin1*, which encode the N-BAR domain involved in membrane curvature, were relatively conserved and characterized by fewer AS events. Despite the large number of detected *Bin1* isoforms, however, almost a third of ONT *Bin1*-mapped reads were annotated to two isoforms: LR.Bin1.99, the known canonical isoform (Bin1-201, ENSMUST00000025239.8) (*n* = 16,624 ONT FL reads, 9.4% of *Bin1* ONT FL reads), and LR.Bin1.101, a novel splicing variant of LR.Bin1.99 with a similar exonic structure (*n* = 36,593 ONT FL reads, 20.7% of *Bin1* ONT FL reads) (Figs. 3D and 7C). While the cortical expression of LR.Bin1.99 was broadly consistent across genotype (Fig. 7D), there was a notable down-regulation at 8 months in TG mice vs WT mice that was paralleled by the significant upregulation LR.Bin1.224, another known isoform (Bin1-205, ENSMUST00000234496.1) (log2FC = 1.19,

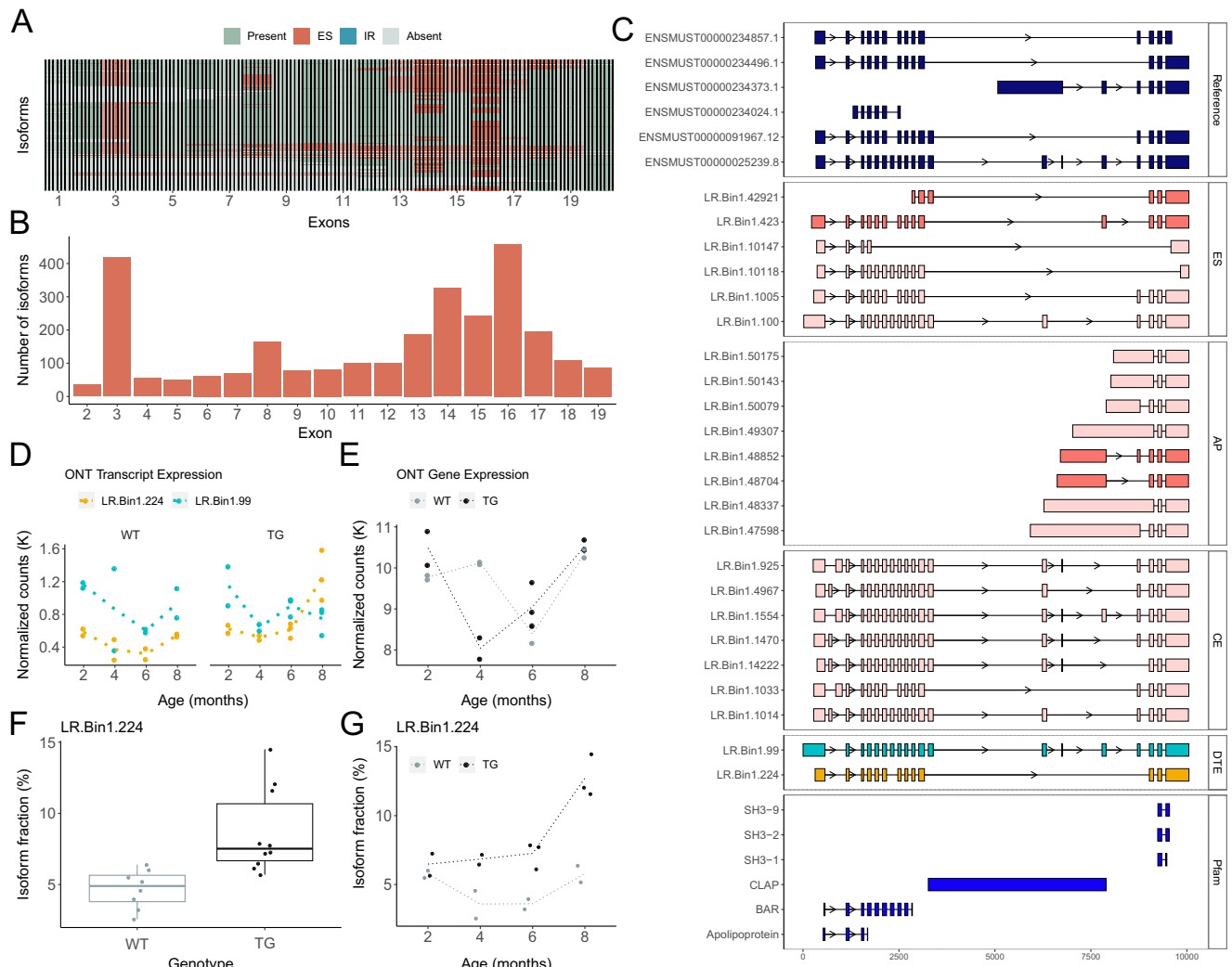

**Fig. 7 | Isoform switching in *Bin1* associated with tau pathology in TG mice.**
**A** A cluster dendrogram depicting the repertoire of *Bin1* isoforms detected using targeted long-read sequencing in the entorhinal cortex from WT and TG mice. Each row corresponds to a distinct isoform and each column represents a known exon. **B** A bar-plot of the number of isoforms in which a known exon (mm10, GENCODE) is skipped. The numbering of each known exon is determined from flattening the reference transcripts (mm10) using *FICLE*. **C** Visualization of the reference and selected novel isoforms of *Bin1* characterized by exon skipping events (ES), intron retention (IR), alternative promoter use (AP) and novel cryptic exons (CE). Isoforms are coloured by structural category (pink – NIC, light pink – NNC). **D** Transcript

expression, determined from normalized ONT full-length read counts, of two known isoforms: LR.Bin1.99, (Bin1-201, ENSMUST00000025239.8) and LR.Bin1.224 (Bin1-205, ENSMUST00000234496.1). Isoforms are coloured with reference to **C**. **E** *Bin1* gene expression (normalized ONT full-length read counts) in WT (gray) and TG (black) mice across age. Isoform usage (proportion) of LR.Bin1.224 stratified by (**F**) genotype and (**G**) age (*n* = 18). For the shown box-plot, the middle box represents the interquartile range (IQR), the middle line represents the median, and the whisker lines represent the minimum (quartile 1 – 1.5 × IQR) and the maximum (quartile 3 + 1.5 × IQR).

FDR = $4.78 \times 10^{-4}$) (Fig. 7D), which differs from LR.Bin1.99 by additional skipping of exon 7 and exon 16 (Fig. 7C). Notably, we observed differential transcript usage of LR.Bin1.224 between WT and TG mice (WT: mean IF = 4.7%, TG: mean IF = 8.7%, Mann-Whitney-Wilcoxon test: *P* = $1.18 \times 10^{-3}$) (Fig. 7F) and with the progression of tau pathology in TG mice (mean IF in TG at 2 months = 6.5%, mean IF in TG at 8 months = 12.7%) (Fig. 7G), which likely contributes to the overall significant DTU observed in *Bin1* (Supplementary Fig. 11). These findings corroborate a recent study showing differential *BIN1* isoform expression in the temporal lobe associated with AD in humans[29]. The down-regulated mouse LR.Bin1.99 (Bin1-201) isoform shows high homology with the human *BIN1* isoform 1 (ENST00000316724.10, 87.2% homology, 79% query cover) that is significantly down-regulated in AD, whereas the up-regulated mouse LR.Bin1.224 (Bin1-205) isoform shows high homology to the human *BIN1* isoform 9 (ENST00000409400.1, 88.2%

homology, 50% query cover) that is significantly up-regulated in AD (Supplementary Fig. 17). Despite these striking differences in the abundance of specific *Bin1* isoforms with the progression of tau pathology, there was no overall gene-level expression difference of *Bin1* between WT and TG mice (log2FC = −0.079, *P* = 0.65) (Fig. 7E), further highlighting the importance of performing transcript-level analyses.

**Targeted long-read transcript sequencing in human cortex reveals a similar transcript repertoire across AD genes and confirms the upregulation of the dominant TREM2 isoform in AD**
To further explore whether transcript-level expression differences identified in TG mice reflect changes observed in human AD, we performed targeted ONT sequencing of the same 20 genes in post-

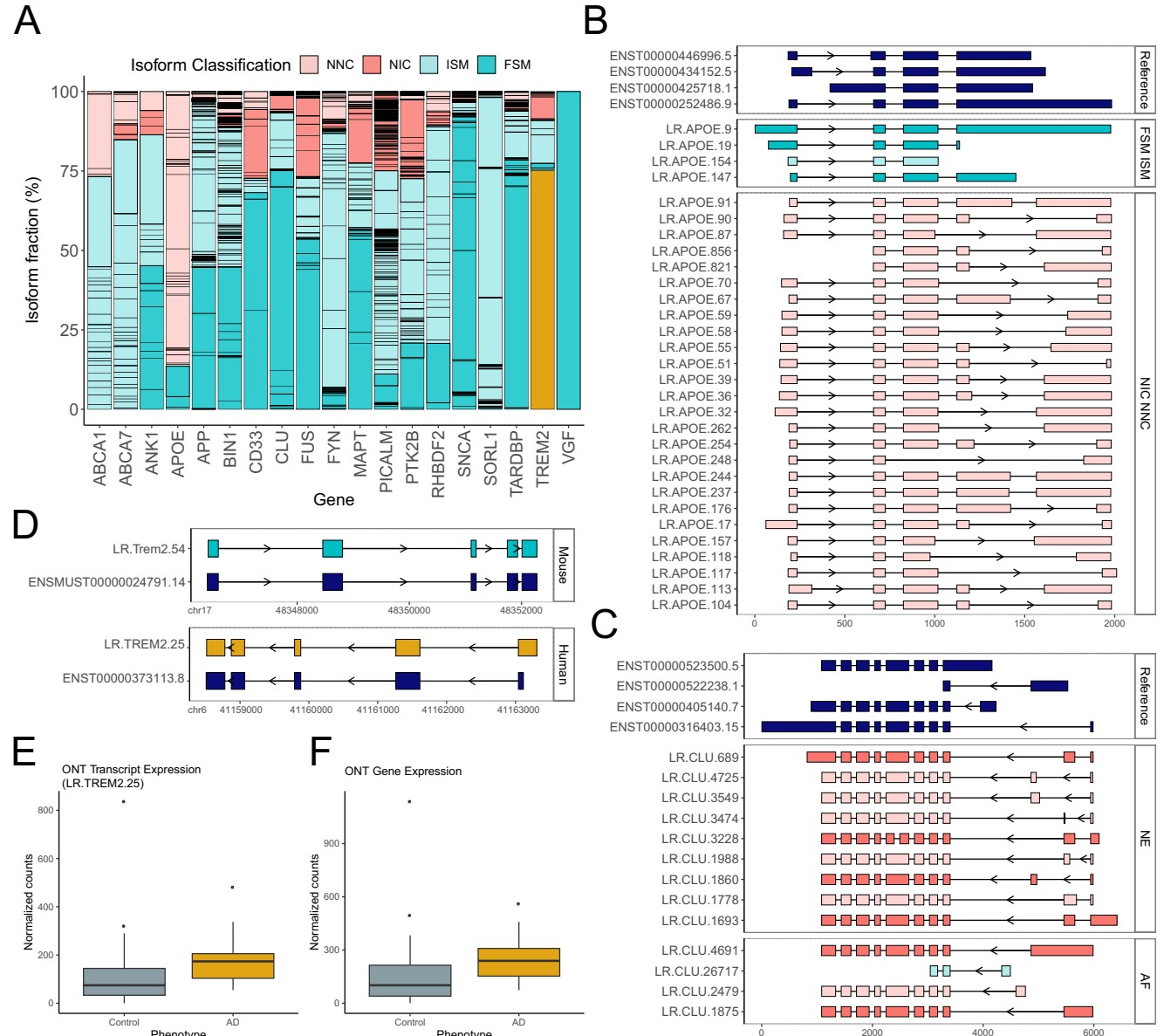

**Fig. 8 | Transcriptional diversity of AD-associated genes in post-mortem human cortex. A** The landscape and usage of different isoforms expressed from each of the 20 targeted AD genes in human cortex, colored by *SQANTI3* structural category (FSM Full Splice Match, ISM Incomplete Splice Match, NIC Novel in Catalog, NNC Novel in Catalog). The lines across the bars delineate individual isoforms. The isoform fraction per gene is determined by dividing the mean full-length read counts of the associated-isoforms across all the samples by the total normalized mean read counts of all the associated isoforms. **B** Visualization of *APOE* isoforms detected in the human cortex, highlighting the complexity of the last exon and 3' UTR (highlighted in green), analogous to that observed in mouse cortex (Fig. 3F). **C** Visualization of a subset of *CLU* isoforms characterized with

alternative first (AF) and novel cryptic exons (CE) (highlighted in green), analogous to those observed in mouse cortex (Fig. 6B). **D** Visualization of the dominant mouse *Trem2* (LR.Trem2.54) isoform and the corresponding human *TREM2* isoform (LR.TREM2.25), which was upregulated in tissue from individuals with and without AD pathology. Box-plot of (**E**) LR.TREM2.25 transcript expression (normalized ONT full-length read counts) and (**F**) overall *TREM2* gene expression (normalized ONT full-length read counts) in human cortex tissue from AD individuals and controls (*n* = 44). The middle box represents the interquartile range (IQR), the middle line represents the median, and the whisker lines represent the minimum (quartile 1 − 1.5 × IQR) and the maximum (quartile 3 + 1.5 × IQR).

mortem cortex tissue dissected from 44 individuals (21 donors with advanced AD neuropathology and 23 neuropathology-free control donors, 50% male, age range = 61 − 101, mean age (SD) = 81.41 (9.3), Supplementary Data 9). Applying the same stringent QC and bioinformatics pipeline (see Methods), we detected 2374 isoforms annotated to the 20 targeted genes, including hundreds of rare novel isoforms (Fig. 8A and Supplementary Data 10) characterized with exonic structures similar to those identified in the mouse cortex. The similarity between mouse and human transcripts for the 20 genes is exemplified by *APOE*, which was characterized by the complex use of

alternative 5' and 3' splice sites in the last exon encoding the apolipoprotein domain (Figs. 8B and 3F), and *CLU* which was characterized by extensive usage of alternative first exons (Fig. 8C) and novel cryptic exons between exon 1 and 2 (Fig. 8C and Supplementary Fig. 9). We observed that the majority of the 20 target genes were dominated by the expression of a few major isoforms that were homologues of the dominant mouse isoforms (Fig. 8A). For example, the human dominant *TREM2* isoform (LR.TREM2.25, ENST00000373113.8) comprised ~75% of *TREM2* transcripts and is homologous to the dominant mouse *Trem2* transcript (LR.Trem2.54/Trem2-201, ENSMUST00000024791.14)

(Fig. 8D). Similarities in expressed transcripts between mouse cortex and human cortex were also observed for *Clu/CLU* (Supplementary Fig. 16) and *Bin1/BIN1* (Supplementary Fig. 17). Although we had limited power to detect DETs associated with AD given the relatively small number of samples (Supplementary Figs. 16 and 17), we found that the dominant *TREM2* transcript (LR.TREM2.25) was significantly upregulated in cortex tissue from AD patients compared to non-AD controls (log2FC = 3.23, Welch's *t*-test: $P = 1.13 \times 10^{-2}$, Fig. 8E), paralleling the association with tau pathology observed in TG mice and driving the overall gene-level increase in *TREM2* gene expression observed in AD (log2FC = 3.31, Welch's *t*-test: $P = 5.72 \times 10^{-3}$, Fig. 8F).

## Discussion

We leveraged the power of long-read sequencing to characterize transcript diversity in the mouse cortex and identify differentially expressed transcripts associated with the progression of tau pathology. Our study represents the first analysis of isoform diversity using long-read cDNA sequencing in a transgenic model of AD pathology and highlights the importance of differential transcript usage, even in the absence of gene-level expression alterations, as a potential mechanism underpinning the consequences of pathogenic tau deposition. Ultra-deep targeted long-read sequencing of genes previously implicated in AD revealed hundreds of novel isoforms driven by AS events and transcript level splicing differences paralleling the development of tau pathology and reflecting changes observed in human AD brain tissue. We also characterized the transcriptional landscape for the same 20 genes in the human AD cortex, highlighting striking similarities with the repertoire of transcripts identified in mice. Our transcript annotations and novel bioinformatics tool (*FICLE*) for the characterization of long-read transcript sequencing data are provided as a resource to the research community.

Our whole transcriptome analysis confirmed findings from our previous analysis of rTg4510 mice using short-read RNA-Seq[5], revealing highly-similar gene-level expression differences associated with tau pathology in TG mice. We were able to leverage our full-length transcript reads, however, to identify the specific transcripts driving these gene-level differences; for example, we confirmed the dramatic upregulation of *Gfap* and showed that this was primarily driven by the increased abundance of the canonical Gfap-201 isoform.

To enable a more comprehensive analysis of alternative splicing and differential transcript use associated with tau pathology, we performed ultra-deep long-read cDNA sequencing on a targeted panel of 20 genes previously implicated in AD using custom-designed biotinylated probes to enrich full-length transcripts prior to both ONT nanopore sequencing and PacBio Iso-Seq. We identified thousands of alternatively-spliced isoforms and detected many novel transcripts annotated to these genes highlighting the power of targeted cDNA sequencing. We established *FICLE* – a tool to handle and document the complexity of these isoforms detected from high-throughput long-read sequencing – which we subsequently used to comprehensively characterize AS events.

Our findings highlight the extent to which AS events contribute to isoform diversity in the cortex and highlight the power of long-read sequencing approaches for transcriptional profiling. The deep sequencing depth achieved using target enrichment allowed us to reliably identify DETs using normalized full-length read counts derived from long-read sequencing. We identified widespread transcriptional variation associated with the progression of tau pathology in TG mice with evidence of altered splicing and transcript expression across the 20 AD genes targeted in our experiments. We identified tau-associated DETs annotated to six of the 20 target genes (*Apoe, App, Cd33, Clu, Fyn,* and *Trem2*). We also identified evidence of DTU associated with tau pathology in genes not characterized by an overall difference in gene expression, highlighting the power of long-read sequencing for characterizing transcriptional differences not detectable using traditional short-read RNA-Seq approaches. Many of the differences observed parallel changes reported in previous studies of AD in the human cortex. For example, we find evidence of differential *Bin1* isoform use in older TG mice paralleling findings from a study in human AD post-mortem cortex[29]. We show that the differentially expressed isoforms primarily differed by the presence/absence of exons encoding the clathrin-binding domain (CLAP) domain, which is involved in endocytosis and is also highly variable among human-equivalent *BIN1* isoforms[27]. Our results further implicate altered exon splicing as a potential mechanism contributing to the role of *Bin1* in tau pathology. Finally, we also characterized transcript diversity in the same 20 AD genes in post-mortem cortex tissue from AD patients and controls, finding a similar isoform landscape to the mouse. Of note, we identified significant upregulation of the dominant *TREM2* isoform in AD cortex, recapitulating the increased expression of the corresponding homologous transcript in TG mice. Our transcript annotations for these 20 genes will have future utility in interpreting findings from genetic studies of AD.

Our results should be interpreted in the context of several limitations. First, because the PacBio Iso-Seq and ONT protocols do not include 5′-cap selection, we cannot exclude the possibility that some of the shorter isoforms identified in our study result from 5′ degradation. By using two long-read sequencing platforms, however, to characterize isoforms in the mouse cortex, in addition to short-read RNA-Seq, we were able to validate a large proportion of our isoform annotations and reduce the number of these artifacts. Of note, many of the novel transcripts identified in this study were rare and excluded from downstream association analyses. Second, our primary analyses were performed on 'bulk' entorhinal cortex tissue, comprising a heterogeneous mix of neurons, oligodendrocytes and other glial cell-types. Despite compelling evidence from recent studies reporting cell-specific transcriptional signatures in disease[30], we were unable to systematically explore these differences in this study. Of note, novel approaches for using long-read sequencing approaches in single cells will enable a more granular approach to exploring transcript diversity in the cortex[31]. Furthermore, while isoform expression was normalized for potential technical cofounders (e.g. library sequencing depth), it was not possible to account for differences in cellular composition between WT and TG mice across all samples. Given neuronal loss and astrogliosis are prominent hallmarks of AD pathogenesis, we were unable to fully discern whether the transcript-level variation associated with tau pathology reflects changes in cell composition associated with pathology. In an attempt to address this issue, we performed long-read transcript sequencing on neuron- and glia-enriched nuclei populations isolated from the bulk cortex from a subset of samples using a FANS protocol developed by our group[12]. Third, we profiled entorhinal cortex tissue given that this is one of the first regions of the brain to be affected by AD pathology. However, tissue-specific differences in splicing and isoform usage of AD-risk genes have been previously reported[32]. Furthermore, we profiled tissue only from female mice to reduce heterogeneity in our analyses. However, a number of sex differences have been reported, with female mice exhibiting earlier and more severe cognitive and behavioral impairments than male TG mice[33]. The insertion of human *MAPT* transgene is also known to contribute to neuropathology by disrupting several endogenous mouse genes[34]. Future work would cross-examine results from our study with transcriptional variation in other mouse models, tissue types and male mice for a more comprehensive understanding of the development of tau pathology in the AD brain. Finally, to enable a more comprehensive analysis of differential splicing in AD, we focused on a panel of 20 AD-associated genes for targeted sequencing. We acknowledge that this is not an exhaustive list of the genes involved in AD, but our targeted approach enabled the ultra-deep profiling of transcripts expressed from selected loci.

In summary, our study identified transcript-level differences in the entorhinal cortex associated with the accumulation of tau pathology in TG mice. Importantly, we identified changes in the abundance of specific transcripts that drive the altered expression of AD genes, with evidence of isoform switching events that could have important functional consequences. Taken together, our results demonstrate the utility of long-read sequencing for transcript-level analyses, facilitating the detection of AS events associated with the development of AD pathology. We have made our transcript annotations (mouse and human cortex) and analysis pipeline freely available as a resource to the community to stimulate further research in this area.

## Methods

### Samples

Mouse entorhinal cortex tissue was dissected from female wild-type and rTg4510 transgenic (TG) age-matched littermate mice in accordance with the UK Animals (Scientific Procedures) Act 1986 and with approval of the local Animal Welfare and Ethical Review Board, granted and approved by Eli Lilly and Company Ltd. rTg4510 mice, licensed from the Mayo Clinic (Jacksonville, FL, USA), were bred on a mixed 129S6/SvEvTac + FVB/NCrl background (heterozygous tau responder x heterozygous tTA effector). Mice were housed under standard conditions (constant temperature and humidity) with a 12 h light/dark cycle in individually ventilated cages (up to 5 animals per cage), with free access to food (Teklad irradiated global rodent diet (Envigo, UK)) and water. Entorhinal cortex tissue was dissected from 29 female rTg4510 TG and 30 female WT mice, aged 2, 4, 6 and 8 months ($n = 7 - 8$ mice per group). rTg4510 mice recapitulate AD tauopathy through the overexpression of the human tau transgene, *MAPT* P301L, which harbors the FTD-associated P301L mutation[35]. It contains four microtubule-binding domains while lacking the N-terminal segment (4R0N), and exons 2 – 3 of the mouse prion protein gene Prnp. The transgene expression is controlled under the calcium calmodulin kinase II promoter (CaMK2a) and is largely restricted to the forebrain[35], with rapid age-dependent spread of neuropathology starting from as early as 2 months in the neocortex to the hippocampus by 5 months. Post-mortem human cortex samples from 44 individuals (21 with advanced AD neuropathology and 23 with minimal neuropathology, 50% male, age range = 61 – 101, mean age (SD) = 81.41 (9.3)) were provided by the Brains for Dementia Research (BDR) Cohort[36]. All samples underwent detailed neuropathological assessment and were stratified by tau pathology using Braak neurofibrillary tau staging[37]. Ethical approval for the study was granted by the University of Exeter Medical School Research Ethics Committee (13/02/009).

### RNA isolation and highly-parallel RNA sequencing

Short-read RNA-Seq data was previously generated for each mouse to identify differentially expressed genes associated with the progression of tau pathology[5]. Briefly, RNA was isolated using the AllPrep DNA/RNA Mini Kit (Qiagen, UK) from ~5 mg tissue and quantified using the Bioanalyzer 2100 (Agilent, UK). RNA-Seq libraries were subsequently prepared with the TruSeq Stranded mRNA Sample Prep Kit (Illumina) and subjected to 125 bp paired-end sequencing using a HiSeq2500 (Illumina). Raw sequencing reads were then processed and mapped to the mouse (mm10) reference genome using *STAR*[38] (v1.9). In our 'hybrid' approach, RNA-Seq-derived gene and transcript expression were then determined by aligning RNA-Seq reads to the whole transcriptome Iso-Seq dataset (*SQANTI3* filtered) generated in this study using *Kallisto*[39] (v0.46.0).

### Whole transcriptome Iso-Seq library preparation and SMRT sequencing

Whole transcriptome PacBio Iso-Seq was performed on RNA isolated from a subset of 12 mice (3 WT and 3 TG at ages 2 and 8 months) (Supplementary Data 1). First strand cDNA synthesis was performed on

200 ng RNA using the SMARTer PCR cDNA Synthesis Kit (Clontech, UK) according to manufacturer's instructions, and oligo(dT) primers that hybridize to the poly(A) tails of mature mRNA molecules (Supplementary Data 12). A total of 14 PCR cycles of amplification was performed for each sample using PrimeSTAR GXL DNA Polymerase (Clontech, UK). Library preparation of the amplified cDNA products was then performed using SMRTbell Template Prep Kit v1.0 (PacBio, USA). Sequencing was performed on the PacBio Sequel 1 M SMRT cell. Samples were processed using either the version 3 chemistry (diffusion loading at 5 pM, 4 h pre-extension, 20 h capture) or version 2.1 chemistry (magbead loading at 50 pM, 2 h pre-extension, 10 h capture).

Raw reads were processed and de-multiplexed with *Iso-Seq* (v3), aligned to the mouse reference genome (mm10, GENCODE) using *pbmm2*[40] (v1.10.0, a *Minimap2* wrapper adapted for PacBio data, parameters: --preset ISOSEQ --sort), collapsed to full-length transcripts using *Iso-Seq collapse*[41] (v3.8.2) and annotated using *SQANTI3* (v5.0) in combination with the mouse reference gene annotations (mm10, GENCODE, vM22). *SQANTI3* filtering was performed using the default JSON file, with the sole modification of relaxing the "min_cov" (the minimum short-read RNA-Seq coverage threshold of non-canonical junctions) from 3 to 0 for non-FSM isoforms. Given the extensive coverage achieved by our targeted long-read sequencing data and the comparably lower depth of our short-read RNA-Seq data for the target genes, we concluded that this parameter was redundant. Full-length Iso-Seq read counts from each sample were extracted from the *Iso-Seq collapse* "read_stat.txt" file with the CCS read ID as sample identifiers. The number of reads sequenced for WT and TG mice is provided in Supplementary Data 11.

### Targeted Iso-Seq library preparation and SMRT sequencing

Targeted PacBio Iso-Seq was performed on RNA isolated from a subset of 24 mice (3 WT and 3 TG at ages 2, 4, 6 and 8 months) (Supplementary Data 1). The same WT and TG mice at aged 2 and 8 months were previously sequenced using whole transcriptome PacBio Iso-Seq (Supplementary Data 1). First-strand cDNA synthesis was performed on 200 ng RNA using the SMARTer PCR cDNA Synthesis Kit (Clontech) with specific oligo(dT) barcodes (Supplementary Data 12) that hybridize to the poly(A) tails of mature mRNA molecules and allow samples to be multiplexed. Large-scale PCR amplification was subsequently performed using 14 cycles, and the resulting amplicons were subjected to targeted enrichment using custom-designed probes (IDT, UK) (Supplementary Data 13). To avoid off-target binding, we manually assessed the list of probes for each target gene under the following criteria: i) all of the exons must be covered by at least one probe, ii) probes should be spaced 300 – 500 bp within each exon (equivalent to 0.2× – 0.3× tiling density), iii) probes with the highest GC content (40 – 65% GC content) and lowest number of blast hits were selected from the contiguous cluster, and iv) any probes covering the intronic regions were removed. Following successful enrichment for target genes, Iso-Seq library preparation was performed using the SMRTbell Template Prep Kit v1.0 (PacBio) for subsequent sequencing on three PacBio Sequel 1 M SMRT cells. Raw reads were processed and de-multiplexed with *Iso-Seq* (v3). The number of reads sequenced for WT and TG mice is also provided in Supplementary Data 11.

### Targeted ONT library preparation and nanopore sequencing

Ultradeep targeted ONT nanopore sequencing was performed on RNA isolated from a subset of 18 mice (8 WT and 10 TG across ages 2, 4, 6 and 8 months). All mice were previously sequenced using targeted PacBio Iso-Seq (Supplementary Data 1). ONT library preparation was undertaken using the Ligation Sequencing Kit (SQK-LSK109, ONT) after target enrichment as described above. Sequencing was subsequently performed on the ONT MinION using two FLO-Min106D flow cells. Raw ONT reads were then basecalled using *Guppy* (v4.0) and

reads with Phred (Q) < 7 were discarded. Primers and ONT adapters were removed using *Porechop* (v0.2.4) to generate full-length reads for each sample. After trimming of poly(A) tails using *Cutadapt* (v2.9) and removal of non-polyA reads, full-length reads were then aligned to the mouse reference genome (mm10, GENCODE) using *Minimap2* (v2.17, parameters: "-ax splice") and corrected using *TranscriptClean* (v2.1). The number of reads sequenced for WT and TG mice is also provided in Supplementary Data 11.

Ultradeep targeted ONT nanopore sequencing was also performed on RNA isolated from post-mortem human cortex tissue (*n* = 44, 21 AD individuals and 23 controls, 50% male, age range = 61 – 101, mean age (SD) = 81.41 (9.3), Supplementary Data 9). Enriching for the same panel of AD-associated genes with human custom-designed probes (Supplementary Data 14), ONT library preparation was undertaken using the SQK-LSK114 ligation sequencing kit and samples were subsequently sequenced on the ONT PromethION using one FLO-PRO114M flow cell. A full protocol of our customized targeted ONT sequencing approach is provided on protocols.io (https://doi.org/10.17504/protocols.io.kqdg3xzwzg25/v1[42]). Raw ONT reads were processed through the same bioinformatics pipeline applied to the mouse targeted experiments, with the exception of alignment to the human reference genome (hg38, GENCODE).

### Fluorescence-activated nuclei sorting of different cell populations from rTg4510 cortex followed by ONT nanopore sequencing

Neuron-enriched and glia-enriched nuclei populations were isolated from cortex dissected from a subset of 8 mice (2 WT and 2 TG at ages 2 and 8 months) using the neural stem cell marker – rabbit monoclonal NeuN antibody conjugated to Alexa Fluor 488 (ab190195, Abcam) at a 1:1000 dilution. Briefly, using a method optimized previously by our group, following tissue homogenization and nuclei purification using sucrose gradient centrifugation we used a FACS Aria III cell sorter (BD Biosciences) to collect populations of NeuN+ (neuron-enriched) and NeuN- (glia-enriched) nuclei. A full protocol detailing each step of our nuclei purification protocol is provided at protocols.io (https://doi.org/10.17504/protocols.io.dm6gpbwndlzp/v1[12]). An example of the gating strategy is provided in Supplementary Fig. 18. Isolated RNA from each population was prepared for sequencing using the SQK-LSK114 ligation sequencing kit and sequencing was performed on the ONT PromethION across two FLO-PRO114M flow cells. Raw ONT reads were then processed through the same ONT bioinformatics pipeline as described above.

### Immunohistochemistry to stain for Trem2, tau and microglia in rTg4510 cortex

The right hemisphere of the rTg4510 mice were sectioned using a Tissue TEK VIP processor and embedded in paraffin wax. 6 μm serial sagittal sections (from bregma 0.84 to 3.00) were obtained using rotary microtomes (HM 200 from Ergostar and HM 355 S from Thermo Scientific), with sections mounted on glass slides[5]. Using slides sectioned from subset of 8 mice (2 WT and 2 TG at ages 2 and 8 months) overlapping with those profiled using ONT targeted sequencing (Supplementary Data 1), deparaffinization of the tissue was achieved using Histoclear II (HS-202, National Diagnostics), followed by 1 min each in 100%, 90%, 70% Ethanol and 100% Methanol then 5 min in deionised water to rehydrate the tissue sections. Heat induced epitope retrieval was performed using 10 mM Tris Base, 1 mM EDTA, pH9 buffer. Samples were blocked using 5% normal goat serum (S-1000-20, Vector Laboratories) in TBS. To stain for microglia, we used a rabbit recombinant monoclonal Iba1 antibody (ab221790, Abcam) at a 1:500 dilution incubated overnight at 4 °C and labeled using a goat anti-rabbit IgG secondary antibody conjugated to Alexa Fluor 647 (A21245, Invitrogen) at a 1:400 dilution in Dako REAL Antibody Diluent (S202230-2, Agilent). To stain for Trem2, we used a rabbit recombinant

monoclonal TREM2 antibody (ab305103, Abcam) at a 1:500 dilution incubated overnight at 4 °C, and labeled using Alexa Fluor 555 Tyramide SuperBoost™ Kit, goat anti-rabbit IgG (B40923, Invitrogen) according to manufacturer's instructions. To assess tau pathology, we used Phospho Tau (Ser202, Thr205) monoclonal Antibody (AT8) antibody (Thermo Fisher Scientific, MN1020) at a 1:500 dilution overnight at 4 °C, which recognizes tau phosphorylated at Ser409, and goat anti-mouse IgG secondary antibody conjugated to Alexa Fluor 488 (A10680, Invitrogen) at a 1:400 dilution. Images were acquired using the PhenoImager HT at 20× and 40× magnification. Visualization of the digitized tissue sections and delineation of the hippocampus were achieved using the HALO software (v3.6, Indica Labs).

### Merged annotation and quantification from targeted long-read sequencing datasets

To comprehensively characterize the transcripts annotated to each of the 20 target genes, full-length reads from both targeted Iso-Seq (after *Iso-Seq3* cluster) and ONT mouse datasets (after correction with *TranscriptClean*) were merged. The merged dataset was then aligned to the mouse reference genome (mm10, GENCODE, vM22) using *pbmm2* [40] (v1.10.0) and collapsed to full-length transcripts using *Iso-Seq collapse* [41] (v3.8.2). Full-length Iso-Seq and ONT read counts from each sample were extracted using custom scripts that determined the source of the reads clustered from the *Iso-Seq collapse* "read_stat.txt" output file. The merged dataset was then annotated using *SQANTI3* in combination with the mouse reference gene annotations (mm10, GENCODE, vm22), RNA-Seq data from an extended set of samples (*n* = 29 TG, 30 WT) and FANTOM5 CAGE Peaks. *SQANTI3* filtering was performed using the default JSON file, with the sole modification being the decrease of "min_cov" from 3 to 0 for non-FSM isoforms. Isoforms associated with the 20 target genes were subsequently selected and retained if observed more than 10 times (i.e. full-length read count ≥ 10) across any five samples (Supplementary Fig. 4).

### Characterisation of AS events and transcript visualization

We developed a novel python-based tool, *FICLE* (v1.1.2), to accurately assess the occurrence of alternative splicing events by comparing splice sites (exon) coordinates between long-read-derived transcripts and reference transcripts (GENCODE). Common alternative splicing events including alternative first exon use (AF), alternative last exon use (AL), alternative 5' splice sites (A5), alternative 3' splice sites (A3), intron retention (IR) and exon skipping (ES) were assessed. Alternative 5' and 3' splice sites were defined as splice sites differing by more than 10 bp from the known splice site. Other regulatory mechanisms such as alternative transcription initiation (defined by an alternative transcription start site) with the presence of an alternative promoter (AP) and termination (defined by an alternative transcription termination site) with the presence of an alternative terminator (AT), and the presence of novel exons not present in existing transcript annotations, were also evaluated. Open reading frames were predicted using the Coding-Potential Assessment Tool[43] (CPAT) (v3.0.2) with default parameters, and transcripts with coding potential score > 0.44 (recommended threshold for mouse) were predicted as protein-coding. Isoforms were visualized using either the UCSC genome browser or *ggtranscript* (v0.99.9)[44].

### Differential expression and splicing analyses

Differential expression analysis was performed using *DESeq2* [45] (v1.26.0) with Iso-Seq and ONT full-length read counts as proxies of gene and transcript expression. Notably, gene expression was aggregated as the summation of full-length read counts from all isoforms associated with target genes, prior to filtering for rare (lowly-expressed) transcripts (minimum 10 reads across any 5 samples). Briefly, *DESeq2* normalizes read counts using the median of ratios method, estimates dispersion, and tests for differential expression using

negative binomial generalized linear models. Datasets were filtered for lowly expressed isoforms (minimum of 10 reads across all samples). Significant genotype effects between WT and TG mice were identified using the Wald test: ~ genotype. Significant interaction effects to detect progressive changes across age between WT and TG mice (i.e. pathology) were identified using the Wald test to compare the nested regression model: ~ genotype + age + genotype * age. Significance was defined by FDR < 0.05, after adjusting *P values* for multiple testing using the false discovery rate (FDR) method (also known as Benjamini and Hochberg correction). Three human samples (Sample 12, 16 and 27, Supplementary Data 9) were excluded from differential expression analysis due to low sequencing coverage. Because we used tissue from an equal number of male and female donors and our sample size was not powered to differentiate sex-effects, we did not include sex as a covariate in our analyses of human tissue.

Differential transcript usage analysis was performed using *EdgeR spliceVariant*[46] (v3.28.1) within *tappAS*[47] (v1.0.7). Lowly-expressed isoforms were filtered using *minorFoldfilterTappas*, whereby an isoform was only retained if its relative proportion to the major isoform was above a specified fold change (FC) threshold (default FC = 0.5). Adopting *tappAS* DTU analysis and metrics, we were able to quantify the magnitude of redistribution of gene expression across its isoforms between conditions (total change) and detect major isoform switching events (podium change). To illustrate DTU at an isoform and sample level (for example, Fig. 5F, G), the isoform usage was determined by dividing the normalized read count for each associated isoform by the total normalized read counts of all the associated isoforms. To illustrate DTU at an isoform level only (Fig. 3D), the isoform usage was determined by dividing the mean normalized read count for each associated isoform across all the samples by the total normalized read counts of all the associated isoforms.

### Long-read proteogenomics analysis
We used an established long-read proteogenomics pipeline (v1.0.0) to generate a comprehensive protein database for the panel of AD-associated genes targeted in our study[18]. The predicted set of protein isoforms were inferred from identifying different ORFs using CPAT, and transcripts that were predicted to produce the same protein were collapsed into one entry with the most abundant transcript as the representative RNA isoform.

### Reporting summary
Further information on research design is available in the Nature Portfolio Reporting Summary linked to this article.

## Data availability
Raw ONT and PacBio Iso-Seq data generated in this study have been deposited in the Sequence Read Archive (SRA) database (https://www.ncbi.nlm.nih.gov/sra) under accession numbers PRJNA981131 (rTg4510 targeted ONT and PacBio Iso-Seq data), PRJNA663877 (rTg4510 whole transcriptome Iso-Seq data) and PRJNA1085642 (AD post-mortem brain targeted ONT data). The processed intermediate data and UCSC genome browser tracks (merged mouse targeted data, mouse whole transcriptome Iso-Seq data, mouse sorted data and human AD targeted data) are available for download at: https://zenodo.org/doi/10.5281/zenodo.8101907.

## Code availability
All original code supporting this study is available at https://github.com/SziKayLeung/rTg4510[48] (https://doi.org/10.5281/ZENODO.12191598), and the *FICLE* package is available to download at https://github.com/SziKayLeung/FICLE. Detailed protocols for FANS and our modified ONT library preparation approach are available on protocols.io (https://doi.org/10.17504/protocols.io.dm6gpbwndlzp/v1; https://doi.org/10.17504/protocols.io.kqdg3xzwzg25/v).

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

## Acknowledgements

S.K.L. was supported by a UK Medical Research Council (MRC) CASE PhD studentship. This work was funded in part through the MRC Proximity to Discovery: Industry Engagement Fund (Precision Medicine Exeter Innovation Platform reference MC_PC_14127) and a research grant from Alzheimer's Research UK (ARUK-PG2018B-016). I.C. 's doctoral studentship was supported by the Alzheimer's Society in partnership with the Garfield Weston Foundation (grant number: 231). Long-read sequencing data generated in the Brains for Dementia Research (BDR) cohort were supported by the Alzheimer's Society and Alzheimer's Research UK (ARUK), a MRC grant to J.M. (MR/W004984/1) and the National Institute for Health and Care Research (NIHR) Exeter Biomedical Research Centre (BRC). The views expressed are those of the author(s) and not necessarily those of the NIHR or the Department

of Health and Social Care. Development of *FICLE* was supported by a grant from the Simons Foundation for Autism Research (SFARI) (grant number: 573312, awarded to J.M.). Immunohistochemistry experiments were further funded by an ARUK grant awarded to S.K.L. (122314R). Sequencing and computational facilities were supported by an MRC Clinical Infrastructure award (MR/M008924/1); the Wellcome Trust Institutional Strategic Support Fund (WT097835MF); a Wellcome Trust Multi-User Equipment Award (WT101650MA); a Biotechnology and Biological Sciences Research Council (BBSRC) Longer and Larger (LoLa) award (BB/K003240/1) and the University of Exeter High-Performance Computing (HPC cluster). We gratefully acknowledge the help of Dr. Adam Smith and Dr. Gregory Wheildon for the sample preparation and RNA extraction of human post-mortem brain tissue. Figure 1 was Created with BioRender.com released under a Creative Commons Attribution-NonCommercial-NoDerivs 4.0 International license.

## Author contributions

J.M. and S.K.L. designed the study. S.K.L., R.B. and A.R.J. conducted long-read sequencing experiments. A.R.J., I.C. and K.M. conducted short-read RNA-Seq experiments. B.C. and C.F. performed FANS and immunohistochemistry staining respectively. J.H. and S.R. advised on experimental design and aspects of immunohistochemistry. K.M. advised on library preparation and aspects of sequencing. Z.A. provided mouse cortex tissue. J.M., E.H. and E.L.D. obtained funding. S.K.L. undertook primary data analyses and bioinformatics, with analytical and computational input from A.R.J., P.O. and E.H. R.B., Z.A., J.T.B. and E.L.D. helped interpret the results. S.K.L. and J.M. drafted the manuscript. All authors read and approved the final submission.

## Competing interests

Z.A. was a full-time employee of Eli Lilly & Company Ltd at the time this work was performed. The other authors declare no competing interests.
