## [Peer Review File · Nature Communications]

Long-read transcript sequencing identifies differential isoform expression in the entorhinal cortex in a transgenic model of tau pathologyREVIEWER COMMENTS

Reviewer #1 (Remarks to the Author):

This manuscript describes long-read sequencing of selected genes relevant to Alzheimer's disease pathology in a widely used model of tau-associated disease mechanisms, rTg4510. The work comes from a respected group. Strengths of the manuscript are that it employs two long-read sequencing platforms, allowing performance characteristics to be compared, and that the authors present a new informatics tool to support researchers with similar analyses. Specific descriptions of transcripts detected highlight alternative splicing events in several of the genes interrogated. However, as it stands, the work is more an illustration of use of the tool and methods. The paper does not provide new insights into the sequencing methods, it does not explore potential mechanisms driving the apparent relative selectivity of alternative splicing and it does not establish the functional impact of the splicing changes. It thus contributes useful, but relatively incremental information.

More specific points are described below:

1. There is a relatively long history of descriptions of alternative splicing events in AD in targeted genes (e.g., Twine et al PLOS 2011). Previous work in *Drosophila* and in humans has previously established an association between cryptic splicing errors and tau pathology and has progressed further mechanistically to implicated spliceosome disruption (Hsieh et al. Cell 2019). With more comprehensive identification of events, particularly given that it was done in an animal model, it would have been useful to consider this or alternative proposed mechanisms.
2. The broader characterisation of transcripts here highlights the potential to begin to understand factors influencing specific classes or genes for which this occurs. It appeared as though the set of transcripts studied was somewhat more arbitrarily selected for disease "relevance", however. Can any conclusions be drawn concerning why events happen more frequently with one rather than another.
3. A challenge is to develop evidence of the independent pathogenic significance of alternative splicing, particularly given the relatively lower frequencies of such events. Development and testing of a hypothesis regarding this is needed. Can the authors extend their work in one instance to explore this?
4. Finally, this work needs extension to human tissue to define its relevance. There are interpretative changes with this model. Disruption of the endogenous mouse gene with random insertion of the human variant transgene itself causes pathology (Gamache et al Nat Comm 2019). The mice also express very high levels of tau (>10-fold), which itself could contribute to splicing variants. The use of the female mice alone also begs the question of potential gender differences, as the females develop a more aggressive phenotype than the males (Yue et al. Neurobio Aging 2011).

Reviewer #2 (Remarks to the Author):

The manuscript by Leung et al, "Long-read transcript sequencing identifies differential isoform expression in the entorhinal cortex in a transgenic model of tau pathology," is a first of its kind. It reports long-read sequencing results from a commonly used mouse model to study tau pathology associated with Alzheimer's disease and other tauopathies as compared to age-matched wild type controls. The inclusion of two ages (2 months and 8 month-old) further enabled changes to be identified in correlation to disease progression.

The paper is well written with clearly presented methods, interpretations and a balanced discussion. The authors also developed a novel analysis tool for the analyses (FICLE) which they have made available as a resource to the community.

The methods and data are exciting, but the relevance remains to be determined. As pointed out in the discussion, it remains unknown whether differences in cell type abundances between genotypes account for differential gene usage (e.g., more neurons in WT with more glia in rTg4510). Long-read sequencing on pure cell populations (e.g., neurons, microglia, astrocytes, etc.) would be important experiment to help provide biological context to the findings.

Also, what is the functional implications of these AS events? Are any of the transcripts predicted to be translated into functional proteins?

Consider moving *Mapt* from Supplementary Figure 8 (*Mapt*) to the main text. It is of this reviewer's opinion that differences in endogenous *Mapt* AS due to the expression of the human tau transgene is one of the most intriguing findings of the manuscript.

The rationale for the chosen 20 genes for ultra-deep targeted long-read sequencing based on their implications in AD and other neurodegenerative disorders is clear. However, including genes that are traditionally used as internal controls/standards would help the interpretations and perspectives on the observed changes of the 20 genes expected to differ. Some suggested genes to include: *LAS1*, *RRP1*, *GUSB*, and *POLR2B* from PMID: 36757952; *CYC1* and *TBP* from PMID: 30406345; *XPNPEP1* from PMID: 22864814. Or perhaps (even better) you choose a few stable transcripts identified in your short-read paper: PMID: 32049030.

Reviewer #3 (Remarks to the Author):

In this work, Leung et al. leverage two different long-read sequencing platforms to study the entorhinal cortex of transgenic model of tau pathology and identify isoforms diversity. To do so, they generated the Oxford nanopore and PacBio data of the rTg4510 mouse models which carry the humanized *MAPT* gene with the P301L mutation associated with FTD. They develop new pipelines and analytical tools to study this new data generated in the context of AD. This manuscript has aspects that raise several major questions that the researchers should address:

Introduction: The authors should address more clearly the hypothesis they are testing and the selection of the mouse models and long-read sequencing approaches. In addition, more details relating to the state-of-the-art for alternative splicing and transcriptional dysregulation of tauopathies should be provided, and similarly about the sequencing platform. Currently, much of the attention is devoted to the work done in this manuscript, that perhaps can be communicated in the results section.

The *MAPT* P301L mutation is linked with familial FTD, it is not clear why the authors are referring to FTD instead of AD.

What do the authors refer as 'hybrid' approach?

What are the findings the authors find in the two time points related to the 3R/4R ratio?

Are the authors underpowered to perform transcriptome wide-analyses? What are the power estimates, and how these correlate to the findings being reported?

Did the authors identify any novel isoform?

What was the rationale for the selection of 20 candidate genes? There is a far larger candidate genes to study *MAPT* mutations, and ND genes not included.

GRN and *C9orf72* were not included among the 20 candidate genes, nor addressed in the manuscript. What are the findings from this study for these genes?

Why did the authors merge the ONT and Iso-seq reads? How did they correct for any artifact introduced from each platform? Did each platform provide similar results?

The authors report the identification of a very large number of novel transcripts (81,221) for only 20 genes, but supported by low number of reads.

a- A clear hypothesis to justify this extraordinary abundance is that a more stringent data cleaning and QC should be considered.

b- What are the distribution of mature and precursor RNA among this new isoforms?

c- Do the authors find any evidence of the presence of these transcripts in additional datasets of ND (AD, FTD, mouse models, or human brains?) using short or long reads?

d- Most of these novel transcripts seem to be supported only by the ONT platform. Can the authors identify any additional experiments in which this was observed? And, Is this overabundance of transcripts identified if data from each platform is processed and analyzed without being merged?

e- Data provided from VastDB suggest that enrichment of isoforms might be an artifact for ONT but not for Iso-seq. Is that correct?

Is the number of novel isoform detected correlated to the number of reads per library?

Is the number of novel isoform detected correlated to the gene level of expression?

The upregulation of the dominant Trem2 transcript might be suggesting microgliosis in the TG mouse. Can the authors provide additional independent data (staining for microglia?) to evaluate this possibility and account for this in their models?

Microgliosis might also explain some of the findings related to other genes expressed in microglia (BIN1, CLU, etc.)

In general, no attempt was made to characterize the isoforms at the cell-type-specific level. Can the authors employ a 'hybrid' approach to evaluate single-cell RNA-seq data and also RNA from sorted cells to contextualize their findings among cell types? This is highly important for this study, as all of the genes mentioned are related to astrogliosis and microgliosis, which might be confounding these results.

No public data from other groups is used to replicate the author's findings. Several groups have shared transcriptomic data that can be used in a 'hybrid' approach.

No information is provided for the alternative isoform or gene DE .

We thank each of the reviewers for their detailed and constructive comments on our work and appreciate their suggestions for novel analyses and potential improvements. Where novel figures have been added to the manuscript, we include the relevant Figure number in our responses below. We also include a number of additional figures related to the Reviewer's comments at the end of this document. Revised changes to the manuscript are highlighted in blue.

Reviewer 1

1. There is a relatively long history of descriptions of alternative splicing events in AD in targeted genes (e.g., Twine et al PLOS 2011). Previous work in *Drosophila* and in humans has previously establish an association between cryptic splicing errors and tau pathology and has progressed further mechanistically to implicated spliceosome disruption (Hsieh et al. Cell 2019). With more comprehensive identification of events, particularly given that it was done in an animal model, it would have been useful to consider this or alternative proposed mechanisms.

The Reviewer makes an excellent suggestion which prompted us to look more closely at cryptic splicing events (now included in the revised manuscript - see **Lines 264 – 270** and as **Supplementary Figure 9**). Using the *FICLE* tool developed as part of this work, we identified novel exons which are not present in existing reference transcript annotations and further classified them as being located i) upstream of the first exon, ii) downstream of the last exon, and iii) within the known first and last exons (i.e. “cryptic exons”). Across the 20 targeted neurodegeneration genes, we detected novel transcripts with novel exons annotated to *Apoe* (n = 1), *Bin1* (n = 7), *Clu* (n = 1), *Snca* (n = 1) and *Trem2* (n = 7). The cryptic-exon-containing transcripts annotated to *Apoe* (between exon 3 and exon 4, **Supplementary Figure 9A**) and *Clu* (between exon 1 and exon 2, **Supplementary Figure C**) were in-frame with no premature stop codons, thereby escaping NMD and resulting in a predicted translation of peptides containing the respective apolipoprotein and clusterin domain. In contrast, the cryptic exons in transcripts annotated to *Bin1* (between exon 1 and exon 2, **Supplementary Figure 9B**) and *Trem2* (between exon 1 and exon 2, **Supplementary Figure 9D**), while in-frame, resulted in short ORFs (with low hexamer scores) and were subsequently recognised as “non-coding” transcripts using *CPAT*. Finally, we observed an out-of-frame novel cryptic exon in a novel transcript annotated to *Snca* (between exon 1 and 2, **Supplementary Figure 9E**) within the synuclein domain that introduced a premature termination codon, thereby predicting NMD. Notably all the above transcripts, while very lowly-expressed (< 10 FL reads), were detected in both WT and TG mice. A summation of the normalised expression, however, revealed an increased abundance of these cryptic-exon-containing transcripts in TG vs WT ($\log_2FC = 0.96$, $P = 1.77 \times 10^{-3}$), and associated with progression of tau pathology ($P = 4.43 \times 10^{-2}$) (**Supplementary Figure 9F**). We have added the results of these new analyses to the revised manuscript.

2. The broader characterisation of transcripts here highlights the potential to begin to understand factors influencing specific classes or genes for which this occurs. It appeared as though the set of transcripts studied was somewhat more arbitrarily selected for disease

“relevance”, however. Can any conclusions be drawn concerning why events happen more frequently with one rather than another.

Our study took an initial transcriptome-wide and then focused on a set of selected target genes. Although a transcriptome-wide approach enables the unbiased analysis of all expressed genes, the targeted approach meant we could obtain ultra-deep coverage of transcripts expressed from the selected loci. Previous studies using targeted long-read sequencing have only focussed on one or a few genes¹⁻³. To our knowledge, this is the first long-read transcriptome study to enrich and target a large panel of genes with a focus on 20 genes previously implicated in AD and other neurodegenerative disorders; as Reviewer 2 notes, “...the rationale for the chosen 20 genes for ultra-deep targeted long-read sequencing based on their implications in AD and other neurodegenerative disorders is clear..”. Using *FICLE* to characterize alternative splicing events across our gene panel, we found a strong significant positive correlation between the total number of splicing events and gene expression ($r = 0.87$, $P = 1.7 \times 10^{-6}$), and no significant correlation with the transcript length ($r = -0.37$, $P = 0.12$), number of exons ($r = -0.34$, $P = 0.15$) or gene length ($r = 0.05$, $P = 0.83$) (**Rebuttal Figure 1**). This suggests that the isoform diversity and splicing frequency in our dataset is primarily associated with gene expression level and presumably biological function. An example of this discussed in the manuscript is *ApoE* (**Figure 3F**), which was the most abundant and isoformic gene despite only containing 5 exons.

3. A challenge is to develop evidence of the independent pathogenic significance of alternative splicing, particularly given the relatively lower frequencies of such events. Development and testing of a hypothesis regarding this is needed. Can the authors extend their work in one instance to explore this?

The Reviewer’s comment prompted us to focus on the abundance of specific AS events associated with tau pathology, testing the hypothesis that these would be more abundant in TG mice. Recent studies have found an increase in transcripts with intron retention (IR) in AD⁴ and tau-induced deficit of nonsense-mediated decay (NMD)⁵. Although there was no overall difference in the expression of IR-annotated and NMD-predicted transcripts between TG and WT mice, there was a significant increase in the abundance of *Trem2* ($\log_2FC = 1.92$, $P = 5.05 \times 10^{-4}$) and *Cd33* ($\log_2FC = 1.19$, $P = 1.66 \times 10^{-2}$) transcripts with IR and associated with NMD (*Trem2*: $\log_2FC = 1.58$, $P = 4.81 \times 10^{-5}$; *Cd33*: $\log_2FC = 1.16$, $P = 5.08 \times 10^{-3}$) TG mice. This is now included in the revised manuscript (see **Lines 308 – 322**), and as **Supplementary Figure 12** and **Supplementary Figure 13**.

4. Finally, this work needs extension to human tissue to define its relevance. There are interpretative changes with this model. Disruption of the endogenous mouse gene with random insertion of the human variant transgene itself causes pathology (Gamache et al Nat Comm 2019). The mice also express very high levels of tau (>10-fold), which itself could contribute to splicing variants. The use of the female mice alone also begs the question of potential gender differences, as the females develop a more aggressive phenotype than the males (Yue et al. Neurobio Aging 2011).

The Reviewer makes another excellent point. As with any transgenic model, there are limitations to the rTg4510 model of tau pathology. This is, however, a well-established model of AD pathology that has been widely used⁶, with transgenic mice exhibiting significant

cognitive impairment with reduction in spatial memory retention^{7,8}. To minimise heterogeneity between samples we focused solely on female mice in this study. There was already some discussion of this in the original manuscript, but we have now also included a sentence in the methods. We note that a number of sex differences have been previously reported for these models, with i) females demonstrating elevated and more progressive pathology than males^{9,10} and ii) males showing higher levels of aggression. Our collaborators at Eli Lilly – who provided us the tissue used in this project – work on females specifically for these reasons, and we did not have access to male samples for this project. In the revised manuscript, we have stressed that future work should focus on examining the extent to which the transcriptional profiles identified here are consistent between male and female mice (see **Lines 548 – 554**).

The Reviewer's comments prompted us to further address the biological relevance of our findings by generating a parallel long-read transcript sequencing dataset on post-mortem dorsolateral prefrontal cortex tissue from 44 human donors (comprising 21 individuals with advanced AD neuropathology and 23 neuropathology-free control individuals). We designed a panel of probes enabling us to target the same panel of 20 target genes using ultra-deep ONT transcript sequencing (see **Figure 1A**). To our knowledge this represents the most thorough analysis of alternative splicing for these genes in human AD.

Importantly, we observed multiple parallels between transcript-level differences in rTg4510 TG mice and our new human dataset (see added **Figure 8**). First, ultradeep targeted sequencing of the human cortex revealed hundreds of rare novel isoforms ($n = 2,374$ isoforms across 20 genes after stringent QC and filtering) with complex patterns of AS events. Our data represents the most systematic examination of transcript diversity for these 20 genes in the human cortex yet undertaken. This is exemplified in *APOE* with complex usage of the alternative 5' and 3' splice sites in the last exon encoding the apolipoprotein (see **Figure 8B**), as seen in our mouse dataset (**Figure 3F**). We also detected a similar *CLU* splicing pattern with extensive usage of alternative first exons and novel cryptic exons between exon 1 and 2 (see **Figure 8C**).

Second, we observed a similar *TREM2* isoform landscape in our human dataset: LR.TREM2.25 (ENST00000373113.8) – the homologue of the dominant mouse LR.Trem2.54 (Trem2-201, ENSMUST00000024791.14) (**Figure 8D**) – emerged as the dominant isoform constituting ~75% of the landscape (**Figure 8A**) corroborating previous studies. Parallel to the mouse findings, LR.TREM2.25 was significantly upregulated in AD ($\log_2FC = 3.23$, Welch Two Sample t-test: $P = 1.13 \times 10^{-2}$, **Figure 8E**) and is likely to also be the primary contributor to the gene-level increase in *TREM2* gene expression ($\log_2FC = 3.31$, Welch Two Sample t-test: $P = 5.72 \times 10^{-3}$, **Figure 8F**). Finally, we observed a similar *CLU* and *BIN1* isoform landscape in mouse and human, with the homologue of the dominant mouse isoform also emerging as the dominant isoform in the human dataset (now included in the revised manuscript as **Supplementary Figure 16** and **Supplementary Figure 17**).

Reviewer 2

The manuscript by Leung et al, “Long-read transcript sequencing identifies differential isoform expression in the entorhinal cortex in a transgenic model of tau pathology,” is a first of its kind. It reports long-read sequencing results from a commonly used mouse model to study tau pathology associated with Alzheimer’s disease and other tauopathies as compared to age-matched wild type controls. The inclusion of two ages (2 months and 8 month-old) further enabled changes to be identified in correlation to disease progression.

The paper is well written with clearly presented methods, interpretations and a balanced discussion. The authors also developed a novel analysis tool for the analyses (FICLE) which they have made available as a resource to the community.

We thank the Reviewer for their positive feedback on our work.

The methods and data are exciting, but the relevance remains to be determined. As pointed out in the discussion, it remains unknown whether differences in cell type abundances between genotypes account for differential gene usage (e.g., more neurons in WT with more glia in rTg4510). Long-read sequencing on pure cell populations (e.g., neurons, microglia, astrocytes, etc.) would be important experiment to help provide biological context to the findings.

We agree that one potential limitation of our analyses is that we profiled bulk cortex tissue. While this would not impact upon our characterisation of transcripts present, the alterations in cell-proportions associated with increased pathology could impact our analysis of differentially expressed transcripts (DETs), and we have expanded our discussion of this limitation in the manuscript (see **Lines 539 – 544**).

Although it is unfeasible to profile cell-specific isoform expression in each of the samples used in this study, the Reviewer’s comment prompted us to undertake additional experimental work to assign transcripts of our targeted genes to specific cell populations. We used a fluorescence-activated nuclei sorting (FANS) protocol developed by our group¹¹ to purify the neuronal and glial nuclei population prior to ONT long-read cDNA sequencing. In line with our key findings, we observed significant upregulation of the dominant *Gfap* (now included as **Figure 2F**) and *Trem2* isoform (now included as **Figure 5D**) in TG vs WT mice, and they were primarily expressed in the glial nuclei populations. Immunohistochemistry staining for *Trem2*, microglia and tau showed that upregulated *Trem2* was localised to activated microglia in close contact with phosphorylated tau in aged TG mice (now included as **Figure 5E, Supplementary Figure 14**). We have elaborated these findings in the revised manuscript – see **Lines 137 – 146** and **Lines 337 – 345**.

Also, what are the functional implications of these AS events? Are any of the transcripts predicted to be translated into functional proteins?

This is an important question by the Reviewer. Following this comment, we have applied an established long-read proteogenomics pipeline¹² to generate a comprehensive protein database for the panel of AD-associated genes in our targeted dataset. Briefly, the predicted set of protein isoforms were inferred from identifying different open reading frames (ORFs) using CPAT, and transcripts that were predicted to produce the same protein were collapsed into one entry with the most abundant transcript as the representative RNA isoform. This

analysis is now included in the **Method** section in the revised manuscript (see **Lines 955 - 960**). Across the 7,162 transcripts annotated to the 20 AD-associated genes, 5,596 (78.1%) transcripts were predicted to be protein-coding with stop codons and of these, ~70% were predicted to have a different ORF resulting in 3,841 unique protein isoforms after *SQANTI Protein* classification and filtering (**Supplementary Figure 6**). The majority of the predicted protein isoforms were classified as pNNC (protein Novel Not in Catalog) (n = 3,044, 79.2%), containing at least one novel element (i.e. a novel N-terminus or splice junction) and being derived from NNC transcripts (**Supplementary Table 5**). These findings are also included in the revised manuscript (see **Lines 234 – 246**).

Consider moving *Mapt* from Supplementary Figure 8 (*Mapt*) to the main text. It is of this reviewer's opinion that differences in endogenous *Mapt* AS due to the expression of the human tau transgene is one of the most intriguing findings of the manuscript.

We thank the Reviewer for this comment and have moved the figure showing upregulated expression of endogenous *Mapt* from Supplementary Figure to Main Figure (see **Figure 4B**). We agree this is an interesting result.

The rationale for the chosen 20 genes for ultra-deep targeted long-read sequencing based on their implications in AD and other neurodegenerative disorders is clear. However, including genes that are traditionally used as internal controls/standards would help the interpretations and perspectives on the observed changes of the 20 genes expected to differ. Some suggested genes include: *LAS1*, *RRP1*, *GUSB*, and *POLR2B* from PMID: 36757952; *CYC1* and *TBP* from PMID: 30406345; *XPNPEP1* from PMID: 22864814. Or perhaps (even better) you choose a few stable transcripts identified in your short-read paper: PMID: 32049030.

This is a good suggestion by the Reviewer. We were able to explore the expression of the genes suggested by the Reviewer using our whole Iso-Seq transcriptome dataset, finding no significant difference in gene-level expression between TG and WT mice, or the expression of specific transcripts of *Las1l*, *Rrp1*, *Gusb*, *Polr2b*, *Cyc1*, *Tbp* and *Xpnpep1*, in addition to a panel of widely-used housekeeping genes (*Gapdh*, *Actb*, *Rrp13a*, *Sdha*) (see **Rebuttal Figure 2**). This confirms the results from our short-read RNA-seq analysis previously undertaken¹³.

Reviewer 3

In this work, Leung et al. leverage two different long-read sequencing platforms to study the entorhinal cortex of transgenic model of tau pathology and identify isoforms diversity. To do so, they generated the Oxford nanopore and PacBio data of the rTg4510 mouse models which carry the humanized MAPT gene with the P301L mutation associated with FTD. They develop new pipelines and analytical tools to study this new data generated in the context of AD.

Introduction: The authors should address more clearly the hypothesis they are testing and the selection of the mouse models and long-read sequencing approaches. In addition, more details relating to the state-of-the-art for alternative splicing and transcriptional dysregulation of tauopathies should be provided, and similarly about the sequencing platform. Currently,

much of the attention is devoted to the work done in this manuscript, that perhaps can be communicated in the results section.

Further details on the current literature surrounding alternative splicing and transcriptional dysregulation of AD tauopathy are now provided in the introduction, along with some additional discussion about the long-read methods used in our study (see **Lines 51 – 59**).

The MAPT P301L mutation is linked with familial FTD, it is not clear why the authors are referring to FTD instead of AD.

While the rTg4510 harbours the FTD-associated P301L mutation, the rTg4510 is a well-established mouse model for recapitulating AD tauopathy. Notably, no causative mutations in MAPT have been identified in AD, though the severity of neurofibrillary tangles has been shown to correlate better with cognitive decline and disease progression than amyloid plaques. We acknowledge that this may not be clear and have provided further details of the mouse model in **Methods** (see **Lines 766 – 778**).

What do the authors refer as 'hybrid' approach?

The 'hybrid' approach refers to our analytic strategy of combining long (Iso-Seq whole transcriptome data, n = 12 samples) and short-read RNA sequencing data (Illumina, n = 59 samples) to perform differential transcript expression analyses. This was described in the manuscript, but we have ensured that our description of the approach is clear where the hybrid annotation is referred (see **Line 639 and 788**).

What are the findings the authors find in the two time points related to the 3R/4R ratio?

The Reviewer is correct that exploring the 3R/4R ratio would be an interesting analysis. Using *FICLE*, we classified mouse *Mapt* transcripts using the splicing pattern of the N-terminus (0N: E2-, E3-; 1N: E2+, E3-; 2N: E2+, E3+) and the microtubule binding repeat domain (3R: E9+, E10-, E11+, E12+; 4R: E9+, E10+, E11+, E12+) (**Rebuttal Figure 3A**). In line with the current literature, the vast majority of *Mapt*-associated transcripts were 4R with only two 3R tau isoforms detected (LR.Mapt.557, LR.Mapt.3042) in WT and TG mice (**Rebuttal Figure 3B**). Although no significant difference in 4R transcript expression was observed between WT and TG mice or across time points, we found a slight increase in the ratio ($P = 0.33$) of mean normalised expression of 4R isoforms to 3R isoforms in TG compared to WT mice (**Rebuttal Figure 3Ci**), with this bias being consistent at both time points (**Rebuttal Figure 3Cii**). Of note, having confirmed the presence of human-specific *MAPT* transgene only in TG mice (**Supplementary Figure 2**), the human *MAPT*-associated transcripts were filtered from downstream analyses. The transgene is known to contain four microtubule-binding domains while lacking the N-terminal segment (4R0N), and exons 2 - 3 of the mouse prion protein gene *Prnp*.

Are the authors underpowered to perform transcriptome wide-analyses? What are the power estimates, and how these correlate to the findings being reported?

We were underpowered to perform transcriptome wide-analyses with our whole transcriptome PacBio dataset, as stated in our second paragraph: "*The constraints on*

sample numbers and sequencing depth associated with the whole transcriptome Iso-Seq approach meant we had relatively limited sensitivity to detect and quantify rare transcripts and identify novel between-group differences using the direct quantification of full-length reads.” This was the rationale for performing ultra-deep long-read cDNA sequencing on the targeted panel of 20 genes. Despite the limited power of our transcriptome-wide analysis, it is reassuring that we could recapitulate gene expression differences identified in our previous RNA-seq study¹³ and attribute these gene-level differences to specific transcripts.

Did the authors identify any novel isoform?

Yes! This is one of the key findings of the study. To reiterate, we identified 77,837 novel transcripts annotated to the 20 genes subjected to ultradeep ONT targeted sequencing dataset. After stringent QC and filtering by expression (choosing a threshold from sensitivity analysis, **Supplementary Figure 4**), our final dataset included 7,162 RNA isoforms comprising 1,000 (14%) known isoforms and 6,162 (86%) novel isoforms. We think this is clearly articulated in the manuscript (see **Line 215** and also in **Abstract**).

What was the rationale for the selection of 20 candidate genes? There is a far larger candidate genes to study MAPT mutations, and ND genes not included.

Our panel of 20 genes for ultradeep targeted sequencing included genes previously implicated in AD including i) genes affected by autosomal dominant mutations associated with neurodegenerative disorders (*App*, *Mapt*, *Snca*, *Fus*, *Tardbp*), ii) genes identified from both AD genome-wide association studies (GWAS) (*Abca1*, *Abca7*, *ApoE*, *Bin1*, *Cd33*, *Clu*, *Picalm*, *Ptk2b*, *Sor11*, and *Trem2*) and epigenome-wide association studies (EWAS) (*Ank1*, *Rhbdf2*), in addition to ii) several genes implicated in other related neurodegenerative phenotypes (*Fyn*, *Trpa1*, *Vgf*). Although we were limited in the number of gene targets that could be included in our enrichment panel in order not to compromise the sequencing coverage per gene, we aimed to include a diverse range of genes. We agree, however, that this is not an exhaustive list of potential genes and have now stressed this limitation more clearly in the discussion (see **Lines 554 – 558**).

GRN and C9orf72 were not included among the 20 candidate genes, nor addressed in the manuscript. What are the findings from this study for these genes?

We agree that both genes are interesting, but we were constrained in the number of genes that could be included in our targeted panel. Given that we did not enrich and sequence *Grn* and *C9orf72* in our targeted datasets and given their limited relevance to tauopathy in the rTg4510 mouse model, we are unable to comment on the findings for these two genes in our targeted dataset. From our whole transcriptome PacBio data, we detected 11 transcripts annotated to *Grn* and 5 transcripts annotated to *C9orf72* (see **Rebuttal Figure 4**). We are currently undertaking a detailed analysis of *C9orf72* isoforms in a cellular model of motor neuron disease, but this is not relevant for the current manuscript.

Why did the authors merge the ONT and Iso-seq reads? How did they correct for any artifact introduced from each platform? Did each platform provide similar results?

The Reviewer is correct – because we performed ONT nanopore sequencing and Iso-Seq on the same set of samples and targeted genes, the reads from both experiments were merged after raw data processing to facilitate methodology comparison and ease downstream processing (i.e. differential expression analyses). Of note, the reads from each sample sequenced using the respective platform can be found in **Supplementary Table 11**. As outlined in the Methods, raw reads from both sequencing approaches were processed separately to correct for potential artefacts introduced from each platform: artificial concatemers were removed in Iso-Seq reads using the standard Iso-Seq bioinformatics pipeline (*Iso-Seq refine*), whereas ONT reads were further corrected using *transcriptClean* (removing mismatches, microindels, and non-canonical splice junctions) given the slightly higher error-rate of ONT nanopore sequencing. A greater number of reads were obtained from ONT sequencing than PacBio Iso-Seq (see **Figure 3C**), reflecting the technological difference between the two platforms – an insert (cDNA sequence of interest) would be sequenced multiple times from multiple polymerase passes in Iso-Seq, whereas the same insert would only be sequenced once following translocation in nanopore sequencing. However, virtually all the transcripts detected in ONT sequencing were also detected in our Iso-Seq dataset as shown in **Figure 3C**.

The authors report the identification of a very large number of novel transcripts for only 20 genes, but supported by low number of reads.

a. A clear hypothesis to justify this extraordinary abundance is that a more stringent data cleaning and QC should be considered.

The 81,221 novel transcripts were reported to highlight the extensive sequencing depth that we achieved using targeted sequencing, suggesting that we have reached saturation for our target genes. We acknowledge that the vast majority of these transcripts, despite having passed stringent filtering using *SQANTI*, are very lowly expressed and may not be biologically relevant. Consequently, we further filtered our dataset using a stringent threshold (minimum 10 reads across any 5 samples, as described in **Supplementary Figure 4**), resulting in a final dataset of 7,162 isoforms. This final dataset was subsequently used for all downstream analyses, as clearly stated in the manuscript: “After stringent filtering of rare transcripts by removing transcripts with < 10 full-length reads across any five samples, our final dataset included 7,162 transcripts isoforms comprising 1,000 (14%) known isoforms and 6,162 (86%) novel isoforms.” We hope this is now clear.

b- What are the distribution of mature and precursor RNA among this new isoforms?

The Reviewer has not understood the approach used in our experiments. We performed cDNA synthesis using oligo(dT) primers that hybridise to the poly(A) tails of mature mRNA molecules. We therefore do not sequence precursor RNA molecules, given that they have not undergone 3' processing and do not have poly(A) tails. This was described in the manuscript, but we have emphasised this point further in the methods (**Lines 797 and 825**).

c- Do the authors find any evidence of the presence of these transcripts in additional datasets of ND (AD, FTD, mouse models, or human brains?) using short or long reads?

As described above in Response to Reviewer 1, we have further performed long-read sequencing using the same approach in i) FANS-purified nuclei populations from different

cell-types in mouse cortex and ii) post-mortem tissue from AD individuals and matched controls (see revised **Figure 1A**). Briefly, using *GffCompare*, we detected the presence of novel transcripts identified in our initial study in sorted nuclei populations and also found evidence for homologues in human post-mortem brain tissue (see **Figure 8D**, **Supplementary Figure 16**, **Supplementary Figure 17**). The majority of novel transcripts were also supported by short-read RNA-Seq data from overlapping mice generated in our previous study¹³.

d- Most of these novel transcripts seem to be supported only by the ONT platform. Can the authors identify any additional experiments in which this was observed? And, Is this overabundance of transcripts identified if data from each platform is processed and analyzed without being merged?

The majority of novel transcripts were supported by short-read RNA-Seq data, by their proximity to CAGE peaks and were characterised with known poly-A motifs. We have added discussion of this in the revised manuscript (see **Lines 216 – 221**). The observation of more novel transcripts in the ONT data reflects the greater read depth, especially as the majority of these novel transcripts are rare. Merging the datasets has no impact on the number of transcripts detected. The rationale behind merging the datasets is to ease downstream processing and comparison and this approach does not remove (or add) any novel transcripts.

e- Data provided from VastDB suggest that enrichment of isoforms might be an artefact for ONT but not for Iso-seq. Is that correct?

The Reviewer is incorrect. We performed stringent filtering of our ONT data (as described earlier), and the difference in the number of transcripts detected between ONT and Iso-Seq is due to the inherent technological differences (as described above) meaning that we have significantly greater read-depth for our ONT experiments with more power to detect rare novel transcripts. Of note, we perform stringent QC and further filter by expression (as described above). The RNA-Seq data from VastDB is generated from diverse vertebrate cell and tissue types, as well as developmental stages. Although it is useful for understanding the general transcriptional landscape, it should not be taken as evidence that the rare transcripts detected by ONT are artefacts given the overwhelming evidence for transcriptional differences between tissues and across development/ageing. Finally, the new human data generated for our revised manuscript (using targeted ONT sequencing) reflects the mouse data, with high homology in transcript structure and abundance.

Is the number of novel isoform detected correlated to the number of reads per library?

In our targeted dataset, we observed a positive correlation for the number of novel isoforms annotated to the AD-associated target genes and the total number of reads (target and off-target) per sample in both sequencing platforms (n = 18 samples; ONT: Pearson's correlation = 0.69, P = 1.44×10^{-3} ; Iso-Seq: Pearson's correlation = 0.81, P = 1.57×10^{-6} , **Rebuttal Figure 5**). However, there was no significant difference in overall reads between WT and TG (Whole Iso-Seq: P = 0.901, Targeted Iso-Seq: P = 0.503, Targeted ONT: P = 0.812), and across age (Whole Iso-Seq: P = 0.398, Targeted Iso-Seq: P = 0.283, Targeted ONT: P = 0.932) (now provided as **Supplementary Table 11**).

Is the number of novel isoform detected correlated to the gene level of expression?

This is a good question. We similarly observed a positive correlation for the number of novel isoforms annotated to AD-associated target genes and the associated gene expression in our datasets (ONT: Pearson's correlation = 0.816, $P = 1.16 \times 10^{-5}$; Iso-Seq: Pearson's correlation = 0.801, $P = 1.68 \times 10^{-5}$) (see **Rebuttal Figure 6**). Of note, the correlation and the genes detected with the highest gene expression/novel isoform frequency are similar across both targeted datasets.

The upregulation of the dominant Trem2 transcript might be suggesting microgliosis in the TG mouse. Can the authors provide additional independent data (staining for microglia?) to evaluate this possibility and account for this in their models? Microgliosis might also explain some of the findings related to other genes expressed in microglia (BIN1, CLU, etc.)

This is an excellent point by the Reviewer. We agree that the upregulation of the *Trem2* isoform could be a reflection of microgliosis in TG mice. We have therefore performed immunohistochemistry staining for microglia, *Trem2* and phosphorylated tau. Using Iba1, a microglia/macrophage-specific calcium-binding protein, we observed an increase in microglia number in TG vs WT mouse and associated with progression of tau pathology (see **Supplementary Figure 14**). Co-staining of *Trem2* showed that upregulation of *Trem2* co-localised with activated microglia in close contact with tau (**Figure 5E**). Our current findings, consistent with our previous study¹³ and the wider literature, substantiates a role (either causal or consequential) for the upregulation of microglial genes in Alzheimer's disease pathology. The results of our long-read transcript sequencing identifies the specific transcript of *Trem2* that contributes most strongly to this upregulation.

Unfortunately our FANS approach (described above) did not enable us to specifically profile transcripts in this cellular population. While microgliosis would not impact upon the overall characterisation of transcripts present across samples, we acknowledge that it could impact our differential transcript expression analysis and have expanded our discussion of this limitation in the manuscript (see **Lines 539 – 544**).

In general, no attempt was made to characterize the isoforms at the cell-type-specific level. Can the authors employ a 'hybrid' approach to evaluate single-cell RNA-seq data and also RNA from sorted cells to contextualize their findings among cell types? This is highly important for this study, as all of the genes mentioned are related to astrogliosis and microgliosis, which might be confounding these results.

We agree that this is an important point and in response, we have generated additional ONT long-read cDNA sequencing data in purified populations from neuron-enriched and glia-enriched nuclei (see **Methods**). In line with our key findings, we observed significant upregulation of the dominant *Gfap* (now included as **Figure 2F**) and *Trem2* isoform (now included as **Figure 5D**) in TG vs WT mice, and they were primarily expressed in the glial-enriched nuclei (NeuN-) populations. Immunohistochemistry staining for Trem2, microglia and tau showed that upregulated Trem2 protein was localised to activated microglia in close contact with phosphorylated tau in aged TG mice (now included as **Figure 5E**, **Supplementary Figure 14**). We have elaborated these findings in the revised manuscript –

see **Lines 137 – 146** and **Lines 337 – 344**. We mention the development of novel methods for single-cell transcript sequencing in the discussion, but this was outside the scope of the current manuscript.

No public data from other groups is used to replicate the author's findings. Several groups have shared transcriptomic data that can be used in a 'hybrid' approach.

Thank you for the reviewer's comment. To our knowledge, this is the first study to perform long-read sequencing in a mouse model of tau pathology. No public long-read sequencing data is therefore available to be used for validation. However, we have stated throughout the manuscript the relevance of our findings in light of the current literature where possible (see **Lines 350, 362, 392, 425 – 431**). Furthermore, as described in the responses above, we have attempted to validate our findings through multiple approaches: i) performing targeted sequencing using both long-read sequencing approaches – ONT nanopore sequencing and PacBio Iso-Seq, ii) validating the splice junctions of our long-read sequencing data using short-read RNA-Seq data generated in the same mice and applying a hybrid strategy for differential expression analysis (as the Reviewer suggests), iii) generating a separate long-read dataset from sorted nuclei populations from the same mice to assess whether specific isoforms are cell-type-specific, iv) immunohistochemistry staining to validate key findings, and v) integrating publicly available datasets (i.e. FANTOM5 CAGE peaks) and polyA motifs to confirm transcription start and end sites.

No information is provided for the alternative isoform or gene DE.

An entire general section and main figure were dedicated to the results from differential transcript and gene expression analysis between WT and TG mice (see section titled: "*Differential transcript expression associated with the progression of tau pathology in TG mice*" and **Figure 4**). We have subsequently elaborated and discussed the key findings in *Trem2*, *Clu* and *Bin1* in the respective sections and figures (**Figures 5 – 7**). We believe this is clearly articulated in the manuscript.

Rebuttal Figures

Figure 1 Relationship between number of AS events and genic features

Figure 2 Housekeeping gene and transcript expression.

Figure 3 Classification of 3R and 4R tau isoforms in rTg4510 mice.

Figure 4 *Grn* and *C9orf72* associated transcripts in the rTg4510 mouse

Figure 5 Relationship between the number of novel isoforms to the number of reads

Figure 6 Relationship between the number of novel isoforms to gene expression

Rebuttal Figure 1: Relationship between the number of AS events and genic features.

Shown are scatter-plots of the number total number of alternative splicing (AS) events (IR, ES, A5'A3) between **(A)** the normalised mean gene expression deduced from our ONT targeted dataset, **(B)** the number of exons (the maximum number of exons determined from the reference transcript), **(C)** the gene length (GENCODE, mm10), and **(D)** the transcript length (determined from the longest reference transcript). Each dot refers to a target gene in our panel. The number of AS events is determined using *FICLE*.

Rebuttal Figure 2: Housekeeping gene and transcript expression.

Shown are **(A)** gene expression levels of housekeeping genes in WT (grey) vs TG (black) mice, and **(B)** transcript-level expression of the top three most abundant transcripts for selected genes in WT vs TG mice across age (2 vs 8 months). Gene and transcript expression is determined from normalized Iso-Seq full length read counts (whole transcriptome Iso-Seq dataset).

Rebuttal Figure 3: Classification of 3R and 4R tau isoforms in rTg4510 mice.

Shown are **(A)** visualizations of **(i)** mouse *Mapt*-associated isoforms and the respective **(ii)** open reading frame, classified by the N-terminus (1st green box) and microtubule binding repeat domains (2nd green box). The isoforms are classified and panelled as 0N3R (E⁻,E³-,E¹⁰-), 0N4R (E⁻,E³-,E¹⁰+), 1N3R (E²+,E³-,E¹⁰-), 1N4R (E²+,E³-,E¹⁰+), and 2N4R (E²+,E³+,E¹⁰+) respectively. Isoforms were classified using *FICLE* and visualized using *ggtranscript*. Exons 2 – 3, 9 – 12 are highlighted in green and Exon 10 is boxed. **(B)** Mean transcript expression (log₁₀ normalized ONT full length read count from targeted ONT dataset) of 3R and 4R tau isoforms between WT and TG mice and across age. **(C)** The ratio of the mean transcript expression of all 4R tau isoforms to the mean transcript expression of all 3R tau isoforms **(i)** between WT and TG mice, and **(ii)** across age. Transcript expression is determined from normalized ONT full length read count (Targeted ONT dataset). Each dot represents a mouse sample.

Rebuttal Figure 4: *Grn* and *C9orf72* associated transcripts in the rTg4510 mouse

Shown are **(A)** visualization tracks of *Grn* transcripts and **(B)** *C9orf72* transcripts detected in the whole transcriptome Iso-Seq dataset and the **(C)** normalized expression of *Grn* transcripts, and **(D)** *C9orf72* transcripts. Transcripts in the tracks are coloured by *SQANTI3* structural category (Blue – FSM (Full Splice Match), Light blue – ISM (Incomplete Splice Match), Red – NIC (Novel in Catalog)).

Rebuttal Figure 5: Relationship between the number of novel isoforms to the number of reads per library in targeted dataset

Shown is a scatter plot of the number of novel isoforms against the total number of reads per sample sequenced in the **(A)** whole transcriptome Iso-Seq dataset (n = 12 samples), and **(B)** targeted Iso-Seq (n = 24 samples) and ONT dataset (n = 18 samples). The novel isoforms shown are the isoforms associated with the target genes (detected prior to filtering for rare (lowly-expressed) transcripts: minimum 10 reads across any 5 samples). The total reads refer to the summation of target and off-target full-length reads. **(C)** Box-plot of the ratio of novel isoforms to total reads for each sample sequenced using two libraries on the two PacBio SMRT cells (Iso Seq) and two ONT flow cells (n = 9 samples for each cell) in the targeted dataset as a direct comparison.

Figure 6: Relationship between the number of novel isoforms to gene expression.

Shown are scatter plots of the mean number of novel AD-associated target isoforms to the mean gene expression in the **(A)** ONT targeted dataset, and **(B)** PacBio Iso-Seq targeted dataset. The gene expression is determined from the summation of the normalized full-length read counts of associated transcripts. Each dot refers to a target gene, and the mean expression and novel isoform frequency are determined from the average across all samples (ONT: n = 18 samples; Iso-Seq: n = 24 samples).

References

1. Clark, M. B. *et al.* Long-read sequencing reveals the complex splicing profile of the psychiatric risk gene CACNA1C in human brain. *Mol. Psychiatry* **25**, 37–47 (2020).
2. Treutlein, B., Gokce, O., Quake, S. R. & Südhof, T. C. Cartography of neurexin alternative splicing mapped by single-molecule long-read mRNA sequencing. *Proc. Natl. Acad. Sci. U. S. A.* **111**, E1291–9 (2014).
3. Tseng, E. *et al.* The Landscape of Transcripts Across Synucleinopathies: New Insights From Long Reads Sequencing Analysis. *Front. Genet.* **10**, 584 (2019).
4. Ong, C.-T. & Adusumalli, S. Increased intron retention is linked to Alzheimer's disease. *Neural Regeneration Res.* **15**, 259–260 (2020).
5. Zuniga, G. *et al.* Tau-induced deficits in nonsense-mediated mRNA decay contribute to neurodegeneration. *Alzheimers. Dement.* **19**, 405–420 (2023).
6. rTg(tauP301L)4510. <https://www.alzforum.org/research-models/rtgtaup301l4510>.
7. Ramsden, M. *et al.* Age-dependent neurofibrillary tangle formation, neuron loss, and memory impairment in a mouse model of human tauopathy (P301L). *J. Neurosci.* **25**, 10637–10647 (2005).
8. Santacruz, K. *et al.* Tau suppression in a neurodegenerative mouse model improves memory function. *Science* **309**, 476–481 (2005).
9. Blackmore, T. *et al.* Tracking progressive pathological and functional decline in the rTg4510 mouse model of tauopathy. *Alzheimers. Res. Ther.* **9**, 77 (2017).
10. Yue, M., Hanna, A., Wilson, J., Roder, H. & Janus, C. Sex difference in pathology and memory decline in rTg4510 mouse model of tauopathy. *Neurobiol. Aging* **32**, 590–603 (2011).
11. Policicchio, S. *et al.* Fluorescence-activated nuclei sorting (FANS) on human post-mortem cortex tissue enabling the isolation of d. (2020).
12. Miller, R. M. *et al.* Enhanced protein isoform characterization through long-read proteogenomics. *Genome Biol.* **23**, 69 (2022).

13. Castanho, I. *et al.* Transcriptional Signatures of Tau and Amyloid Neuropathology. *Cell Rep.* **30**, 2040–2054.e5 (2020).

REVIEWER COMMENTS

Reviewer #1 (Remarks to the Author):

The revised manuscript is improved, although the authors effort to develop evidence for independent pathogenic significance of the alternative splicing was unsatisfying- another association was noted only. However, I appreciate that this is challenging. Nonetheless, the highlight of the area, the observations and software represent significant steps forward. The authors should be congratulated for some very nice work.

There are a couple of points that could be further developed to make the publication more valuable:

The report focuses on characterisation of splicing diversity in genes associated with neurodegenerative disease. This implies potential neuropathological significance. However, it would be critical to understand whether there is an enrichment in isoform diversity in these genes relative to that in other transcripts (not recognised as associated with pathology) expressed to similar relative levels. I could not appreciate this immediately from the new data (Reb Fig 2) added.

The authors report that there was a relationship between the detection of transcript splice diversity and gene expression level. The biological significance of this seems uncertain: could this merely be a consequence of the low frequency of events and detection limits for the method? Can the authors highlight the relationship between transcript expression and isoform diversity in the manuscript and clarify (in follow on to previous question) whether this is different for housekeeping or neurodegenerative pathology associated genes?

Reviewer #2 (Remarks to the Author):

The authors addressed all of my concerns. Congratulations on a nice contribution.

Miranda Orr

Reviewer #3 (Remarks to the Author):

In this revised version of the manuscript, the authors address many of the critiques and questions raised by the 3 reviewers. I consider this new version has improved largely from the previous submission.

It still calls my attention that the investigators did not find any precursor / or non-polyA transcripts. Indeed, many non-polyA reads are captured in short Illumina RNA-seq from poly-A captured libraries from human postmortem brains. The authors are suggesting that there is no leakage of non-polyA transcripts in their libraries which make be unique of their libraries, which should be addressed in more detail. I encourage the authors to compare their sequences to other studies that used long and short reads to sequence polyA-captured RNA-seq.

Reviewer 1

The revised manuscript is improved...the highlight of the area, the observations and software represent significant steps forward. The authors should be congratulated for some very nice work.

We thank the Reviewer for their positive feedback on our revision.

There are a couple of points that could be further developed to make the publication more valuable:

The report focuses on characterisation of splicing diversity in genes associated with neurodegenerative disease. This implies potential neuropathological significance. However, it would be critical to understand whether there is an enrichment in isoform diversity in these genes relative to that in other transcripts (not recognised as associated with pathology) expressed to similar relative levels. I could not appreciate this immediately from the new data (Reb Fig 2) added.

Thank you for your feedback and apologies that this was not fully addressed in our previous Rebuttal. The comment prompted us to explore this further. Using our whole transcriptome Iso-Seq dataset (which was not limited to our targeted gene panel), we compared the diversity of isoforms expressed from the 20 target AD-associated genes and other protein-coding genes with similar gene expression levels (i.e. within the range of gene expression observed from our target genes) in WT and TG mice (see **Rebuttal Figure 1**, below). Interestingly, we found significantly higher isoform diversity in our targeted panel of neurodegeneration genes compared to other protein-coding genes ($P = 6.09 \times 10^{-3}$). Given the limited coverage of the Iso-Seq data we have not included this preliminary finding in the revised manuscript, but we believe it highlights the potential relevance of alternative splicing at these genes in AD pathology.

The authors report that there was a relationship between the detection of transcript splice diversity and gene expression level. The biological significance of this seems uncertain: could this merely be a consequence of the low frequency of events and detection limits for the method? Can the authors highlight the relationship between transcript expression and isoform diversity in the manuscript and clarify (in follow on to previous question) whether this is different for housekeeping or neurodegenerative pathology associated genes?

The reported relationship between the number of transcripts and gene expression in our Rebuttal was confined to our target genes. Expanding this to our whole transcriptome dataset, we still observed a positive correlation for the total number of isoforms annotated to our panel of 20 target AD-associated genes and the associated gene expression (Iso-Seq: Pearson's correlation = 0.63, $P = 3.87 \times 10^{-15}$). Conversely, this was not observed for the housekeeping genes (Iso-Seq: Pearson's correlation = 0.26, $P = 0.24$) which might be expected given the highly regulated expression of these genes. Given that our targeted sequencing reached

saturation it is plausible that the increased splicing associated with increased expression in genes associated with AD is biologically significant and doesn't simply reflect the detection limit of the sequencing experiments.

Reviewer 2

The authors addressed all of my concerns. Congratulations on a nice contribution.

We thank the Reviewer for their valuable comments on our original draft.

Reviewer 3

In this revised version of the manuscript, the authors address many of the critiques and questions raised by the 3 reviewers. I consider this new version has improved largely from the previous submission.

We appreciate this positive feedback on our revised manuscript.

It still calls my attention that the investigators did not find any precursor / or non-polyA transcripts. Indeed, many non-polyA reads are captured in short Illumina RNA-seq from poly-A captured libraries from human postmortem brains. The authors are suggesting that there is no leakage of non-polyA transcripts in their libraries which make be unique of their libraries, which should be addressed in more detail. I encourage the authors to compare their sequences to other studies that used long and short reads to sequence polyA-captured RNA-seq.

We apologize that this was still confusing. To clarify, the use of standard oligo(dT) primers in our experimental protocol – as with other polyA-captured libraries – should preclude the sequencing of precursor/non-polyA transcripts. We additionally performed stringent QC and processing to remove any remaining sequences with non-polyA ends - as recommended by the ONT and PacBio bioinformatics community. In our whole PacBio Iso-Seq dataset prior to this filtering, a mean 1.99% (mean n = 7,450 reads) of full-length reads (after *Lima*, mean n = 374,126 reads) were non-polyA reads (polyA tail less than 20bp, default *Lima* parameter). In our targeted ONT dataset, a mean 6.12% (mean n = 51,609 reads) of reads (after *Porechop*, mean n = 852,295 reads) were non-polyA reads (polyA tail less 60 bp, default *Cutadapt* parameter). These sequences were removed from downstream analyses and this is now further clarified in the **Methods** (Line 851) to explain why these sequences are not observed in our final dataset.

Rebuttal Figure 1: Comparison of isoform diversity of AD-associated genes

Shown is a histogram of the isoform diversity (number of isoforms) associated with our panel of 20 target genes and other protein-coding genes with similar gene expression level in our whole transcriptome dataset.

REVIEWERS' COMMENTS

Reviewer #1 (Remarks to the Author):

The authors understood and described attempts to directly address the questions raised. These data alone seemed unable to conclusively answer the questions, however. I am grateful to the authors for trying.

Overall, while the paper does not contribute major conceptual advances, but does describe an important technical achievement and sets out some of the questions that now need to be pursued. It is a useful contribution to the literature.

Reviewer #1 (Remarks on code availability):

No. I have not reviewed the code but the data seems presented comprehensively.

We thank Reviewer #1 for their positive feedback on our revision.

We have been through the requested editorial changes and addressed all remaining queries.